# AdaReP: Plug-and-Play Acceleration for World Model Predictive Control using Adaptive Re-Planning

## Abstract

We investigate the integration of model predictive control (MPC) with world models for robotic control tasks. Existing MPC solvers often replan at every step or after very few steps, primarily to mitigate the accumulation of world model prediction errors. However, such frequent replanning incurs substantial computational costs – especially when using large, complex world models. In this work, we theoretically characterize the fundamental trade-off between computational efficiency and control performance in MPC. Our analysis reveals how replanning frequency, model prediction error, and local dynamics sensitivity jointly influence MPC performance, as captured by regret bounds. Based on the analysis, we propose AdaReP, a novel adaptive replanning mechanism for MPC that dynamically modulates the replanning frequency based on online estimates of world model prediction error and local dynamics sensitivity. AdaReP is training-free, plug-and-play, and compatible with various world models and MPC solvers. Experiments on the VP2 simulation benchmark across diverse tasks, as well as real-world robotic tasks including door opening and T-block pushing, show that AdaReP achieves substantial reductions in computation, over 80–90% in the real-world settings while maintaining or improving task success rates. Code will be made public.

## 1 Introduction

This paper studies model predictive control (MPC) combined with learned world models for various robotic control tasks (Ding et al., 2024; Campbell et al., 2023; Wu et al., 2024; Tian et al., 2023a; Zhao et al., 2024). Typically, a predictive world model is first trained to forecast future robot and environment states (*e.g.*, images) based on historical states and robot actions. MPC solvers, such as the Cross-Entropy Method (CEM) or Model Predictive Path Integral control (MPPI), then sample multiple action sequences and query the world model to predict their outcomes. Action sequences that are more likely to achieve the specified goals – based on the predicted future states – are selected for execution (De Boer et al., 2005; Anderson & Moore, 2007). Fueled by the powerful pretrained generative models Ho et al. (2020); Rombach et al. (2022), MPC using these generative world models has been applied to a variety of robotic tasks, including manipulation and navigation Du et al. (2023); Yang et al. (2023); Wang et al. (2024).

Despite their effectiveness, MPC with learned world models can suffer from computational inefficiency. As illustrated in Figure 1, to mitigate cumulative prediction errors, many MPC solvers avoid executing full action plans and instead replan at every step or after only a few steps. This frequent replanning greatly increases the number of world model queries, leading to higher computational cost and reduced control frequency due to delays. The use of large, complex world models further exacerbates this issue.

To this end, we propose AdaReP, a novel adaptive replanning mechanism for MPC. Our **key idea** is to dynamically modulate the replanning frequency in MPC. We start by theoretically characterizing the fundamental trade-off between computation efficiency and control performance in MPC. Our analysis shows how the MPC regret bounds, which capture the performance, can be jointly affected by replanning frequency, world model prediction error, and local dynamic sensitivity. We therefore design an algorithm to adjust the replanning frequency based on online estimates of world model pre-

Figure 1: An illustration of the computational efficiency–control performance trade-off in MPC and our ADAREP: Traditional MPC solvers replan frequently to curb cumulative world-model prediction error, driving up computation. By contrast, ADAREP adjusts replanning frequency on-the-fly using online estimates of prediction error and local dynamics sensitivity, cutting computation while preserving control performance.

diction error and local dynamics sensitivity. ADAREP is training-free, plug-and-play, and compatible with various world models and MPC solvers. An overview of ADAREP can be found in Figure 1.

We conduct an extensive evaluation of our method on both simulated and real-world robotic control, ranging from the diverse tasks in VP2 simulation benchmark to real-world door opening and T-block pushing. Overall, when combined with a collection of world models and MPC solvers, ADAREP demonstrates a substantial reduction of computation – over 80-90% – as measured by number of function evaluations (NFEs), while maintaining the task success rate. Our additional analysis further reveals the control scenarios where ADAREP suits better.

To sum up, our contributions are threefold:

- We provide a rigorous regret analysis of MPC replanning strategies, offering theoretical insights into the computation efficiency-control performance trade-off.

- We develop ADAREP, a practical and efficient plug-and-play adaptive replanning algorithm for MPC, designed to enhance computational efficiency while maintaining robust control performance; unlike prior methods that replan every step, ADAREP decides *when* to replan.

- We present experimental validation in both simulated and real-world robotic manipulation tasks, demonstrating the effectiveness and robustness of our proposed adaptive approach compared to canonical MPC solvers.

## 2 PRELIMINARIES

**Model Predictive Control.** We begin by outlining the general formulation for finite-horizon, discrete-time optimal control, which forms the basis for our discussion. We consider problems characterized by potentially time-varying costs, dynamics, and constraints. The objective is to determine state and control trajectories, denoted by $x_{0:T}$ and $u_{0:T-1}$ respectively, that solve the following optimization problem:

$$\min_{x_{0:T}, u_{0:T-1}} \sum_{t=0}^{T-1} f_t(x_t, u_t; \xi_t^*) + F_T(x_T; \xi_T^*) \tag{1}$$

$$\text{s.t. } x_{t+1} = g_t(x_t, u_t; \xi_t^*), \qquad \forall 0 \le t < T,$$
$$s_t(x_t, u_t; \xi_t^*) \le 0, \qquad \forall 0 \le t < T,$$
$$x_0 = x(0).$$

Here, $x_t \in \mathbb{R}^n$ represents the system *state* at time $t$, and $u_t \in \mathbb{R}^m$ is the *control input* or *action*. The function $f_t$ denotes the time-varying *stage cost*, $g_t$ represents the time-varying system *dynamics*, and $s_t$ encapsulates the time-varying *constraints*. Crucially, these functions are parameterized by $\xi_t^*$,

representing unknown ground-truth parameters governing the system's behavior at time $t$. $F_T$ is a terminal cost function, parameterized by $\xi_T^*$, applied to the final state $x_T$. The initial state is given by $x(0)$.

A widely adopted approach for addressing such problems, particularly in online settings where future parameters $\xi_t^*$ are unknown, is *Model Predictive Control (MPC)*. Solving the full-horizon problem equation 1 directly is often impractical, even if an estimate $\xi_t$ of the true parameters $\xi_t^*$ is available. Two primary challenges arise:

1. **Computational Complexity:** Solving the large-scale optimization problem equation 1 can be computationally prohibitive, especially for long horizons $T$.

2. **Model Mismatch & Error Accumulation:** Using an imperfect model (parameterized by $\xi_t$ instead of $\xi_t^*$) over a long horizon can lead to the accumulation of prediction errors, potentially resulting in significant performance degradation or constraint violations.

To mitigate these issues, MPC employs a *receding horizon* strategy. At each time step $t$, given the current state $x_t$ and potentially updated parameter estimates $\xi_{t:\min(t+k,T)-1}$, MPC solves a *Finite-Time Optimal Control Problem (FTOCP)* over a shorter prediction horizon $k$.

**Definition 2.1** (FTOCP). *The Finite-Time Optimal Control Problem (FTOCP) over the horizon $[t_1, t_2]$, initialized at state $z$, using parameters $\xi_{t_1:t_2-1}$, terminal parameter $\zeta_{t_2}$, and terminal cost function $F(\cdot; \cdot)$, seeks to find the minimum cost:*

$$\iota_{t_1}^{t_2}(z, \xi_{t_1:t_2-1}, \zeta_{t_2}; F) \coloneqq \min_{y_{t_1:t_2}, v_{t_1:t_2-1}} \sum_{t=t_1}^{t_2-1} f_t(y_t, v_t; \xi_t) + F(y_{t_2}; \zeta_{t_2}) \tag{2}$$

$$s.t. \ y_{t+1} = g_t(y_t, v_t; \xi_t), \qquad \forall t_1 \le t < t_2,$$
$$s_t(y_t, v_t; \xi_t) \le 0, \qquad \forall t_1 \le t < t_2,$$
$$y_{t_1} = z.$$

*Let $\psi_{t_1}^{t_2}(z, \xi_{t_1:t_2-1}, \zeta_{t_2}; F)$ denote a corresponding optimal trajectory solution $(y_{t_1:t_2}, v_{t_1:t_2-1})$.*

The FTOCP equation 2 is solved at the current time $t$ over the horizon $[t, \min(t+k, T)]$ using the current state $x_t$ as the initial state $z$. From the resulting optimal control sequence $v_{t:\min(t+k,T)-1}$, only the first control action, $u_t = v_t$, is applied to the actual system dynamics $g_t(\cdot, \cdot; \xi_t^*)$. The system transitions to the next state $x_{t+1}$, and the process repeats at time $t+1$. A typical implementation is described in $\mathsf{MPC}_k^1$ (Algorithm 2).

**Measuring Control Performance of MPC.** We evaluate online control algorithms (ALG) by comparing their executed trajectories against the offline optimal trajectory (OPT), which assumes perfect foresight of ground-truth parameters $\xi_{0:T}^*$.

**Definition 2.2** (Trajectories). *Given initial state $x_0$ and parameters $\xi_{0:T}^*$:*

- Executed Trajectory (obtained from ALG): $x_0 \xrightarrow{u_0} \cdots \xrightarrow{u_{T-1}} x_T$, *where $u_t$ is chosen by* ALG *and $x_{t+1} = g_t(x_t, u_t; \xi_t^*)$.*

- Offline Optimal Trajectory (OPT): $x_0^* \xrightarrow{u_0^*} \cdots \xrightarrow{u_{T-1}^*} x_T^*$, *solving equation 1 with known $\xi_{0:T}^*$.*

Our primary performance metric is *dynamic regret* (Li et al., 2020; Gandhi et al., 2021; Dogan et al., 2023; Goel et al., 2019; Fiacco & Ishizuka, 1990), quantifying the cumulative cost difference between the executed and optimal trajectories due to the online nature of the algorithm:

$$\mathrm{Regret}(\mathsf{ALG}) \coloneqq \mathrm{cost}(\mathsf{ALG}) - \mathrm{cost}(\mathsf{OPT}), \tag{3}$$

where $\mathrm{cost}(\cdot)$ is the total trajectory cost calculated using true parameters $\xi_t^*$:

$$\mathrm{cost}(\mathsf{ALG}) \coloneqq \sum_{t=0}^{T-1} f_t(x_t, u_t; \xi_t^*) + F_T(x_T; \xi_T^*) \quad \mathrm{cost}(\mathsf{OPT}) \coloneqq \sum_{t=0}^{T-1} f_t(x_t^*, u_t^*; \xi_t^*) + F_T(x_T^*; \xi_T^*).$$

**Measuring Computation Efficiency of MPC.** To assess computational efficiency, especially crucial when using learned world models, we choose *Number of Function Evaluations (NFEs):* The

total number of predictive world model queries made by the MPC solver (e.g., CEM, MPPI) during one episode ($t = 0$ to $T$). NFE directly reflects the computation cost and potential for acceleration. Using NFE or similar query counts as a measure of computational efficiency is common practice in related fields, including optimization (ten Eikelder & van Amerongen, 2023), generative modeling (Prasad et al., 2024), and large language model planning (Sun et al., 2023).

## 3 METHODOLOGY

The core objective of our methodology is to enhance the computational efficiency of MPC when using learned world models, specifically by minimizing the Number of Function Evaluations (NFE), without significantly compromising control performance, as measured by dynamic regret. More specifically, we aim to minimize the NFE conditioned on the overall regret is no more than ($\varepsilon + \text{Regret}(\text{MPC}_k^1)$)We achieve this by developing adaptive replanning strategies and analyzing their theoretical properties and practical effectiveness.

### 3.1 THEORETICAL ANALYSIS

Standard Model Predictive Control can be computationally demanding. This is particularly true when employing complex predictive models (e.g., learned world models) and sampling-based optimizers (e.g., Model Predictive Path Integral control (MPPI)(Williams et al., 2017), Cross-Entropy Method (CEM)(De Boer et al., 2005)), which may require hundreds or thousands of model queries per control step.

Natural approaches to accelerate the planning process involve reducing the replanning frequency. Two such strategies are considered:

- $\text{MPC}_k^m$: Execute a fixed number, $m \geq 1$, of actions from the computed plan before replanning (Algorithm 3).

- $\text{MPC}_{k,\epsilon}$: Replan only when the system state deviates significantly (by more than a threshold $\epsilon$) from the previously planned trajectory (Algorithm 4).

Intuitively, reducing the replanning frequency may degrade control performance compared to standard $\text{MPC}_k^1$. We aim to characterize this trade-off between computational savings and performance, measured by dynamic regret, both theoretically and empirically.

Our theoretical analysis builds upon perturbation analysis techniques (Shin et al., 2020; Lin et al., 2021; Shin & Zavala, 2021; Xu & Anitescu, 2019; Na & Anitescu, 2022) and adapts the 3-step analytical pipeline proposed by Lin et al. (2022). Detailed derivations are deferred to Appendices C and D, while the main theoretical results on dynamic regret are summarized in Table 1.

Table 1: Overview of theoretical regret bounds. Here, $L = \max_{0 \leq t < T} L_t$ and $L_* = \max_{0 \leq t \leq T} \max_{0 \leq i \leq m-1} \prod_{s=t}^{t+i} L_s$ characterize the sensitivity (Lipschitz constants) of the dynamics over single and multiple steps, respectively. $E$ represent cumulative prediction errors of the underlying model. Note that when $m = 1$ or $\epsilon = 0$, our results recover (Lin et al., 2022). NFEs decreases as $m, \epsilon, \alpha_L$ and $\alpha_\delta$ increases. Full table on the characterization of NFE details can be found in Table 2.

| Algorithm | Reference | Regret Bound |
|:---:|:---:|:---:|
| $\text{MPC}_k^1$ | Lin et al. (2022) | $O\left(\sqrt{L^2 \text{cost}(\text{OPT}) \cdot E} + L^2 E\right)$ |
| $\text{MPC}_k^m$ | Theorem C.4 | $O\left(\sqrt{mL_*^2 \text{cost}(\text{OPT}) \cdot E} + mL_*^2 E\right)$ |
| $\text{MPC}_{k,\epsilon}$ | Theorem C.6 | $O\left(\sqrt{L^2 \text{cost}(\text{OPT}) \cdot (E + \epsilon E + \epsilon^2 T)} + L^2(E + \epsilon E + \epsilon^2 T)\right)$ |
| $\text{MPC}_{AR}$ | Theorem C.7 | $O\left(\sqrt{\text{cost}(\text{OPT})(L^2 E + \frac{\epsilon_0}{\alpha_L^2}(\epsilon_0 + \frac{1}{\alpha_\delta})T)} + L^2 E + \frac{\epsilon_0}{\alpha_L^2}(\epsilon_0 + \frac{1}{\alpha_\delta})T\right)$ |

Our theoretical analysis reveals that the major additional regret in $\text{MPC}_{k,\epsilon}$ is $\epsilon L^2 E + \epsilon^2 L^2 T$, highlighting the tradeoff between computational efficiency and control performance. This suggests choosing more aggressive $\epsilon$ when both prediction error $E$ and dynamics sensitivity $L$ are small.

Theoretically, we can set $\epsilon \propto \exp(-\alpha_L L) \cdot \exp(-\alpha_\delta E)$, leveraging the exponential decay property $x \exp(-\alpha x) \leq \frac{1}{e\alpha}$ to control regret through parameter $\alpha$.

However, fixed strategies are suboptimal in practice since both prediction error and dynamics sensitivity vary significantly over time. We therefore introduce $\mathsf{MPC}_{AR}$ (Algorithm 5), which adaptively adjusts the threshold online, as detailed in the following subsection.

## 3.2 AdaReP: an Adaptive Replanning Mechanism

The core idea of AdaReP is to dynamically adjust the replanning strategy based on real-time performance metrics, specifically model prediction accuracy and local dynamics sensitivity. The goal is to replan less frequently (saving computation) when the model performs well and the system behaves predictably, but increase replanning frequency when prediction errors rise or the system exhibits higher sensitivity. The detailed procedure is implemented in Algorithm 1 and Algorithm 5.

The adaptation relies on metrics computed at each time step. Let $(y_{t:...}, v_{t:...})$ be the plan computed at time step $t$. After applying the first action $u_t = v_t$ and observing the actual next state $x_{t+1}$ resulting from the true dynamics $g_t(\cdot, \cdot; \xi_t^*)$, we calculate:

1. **Prediction Error:** The deviation between the observed state and the state predicted by the model $g_t(\cdot, \cdot; \xi_t)$ used for planning (denoted $y_{t+1}$ in the plan starting from $x_t$).

$$\delta_{t+1}^o = \|x_{t+1} - y_{t+1}\| \ , \tag{4}$$

where $y_{t+1} = g_t(x_t, u_t; \xi_t)$ according to the internal model.

2. **Local Dynamics Sensitivity Estimator:** An estimate of how much the state changes relative to the control input magnitude, which provides an empirical measure of the system's local sensitivity.

$$\widehat{L}_t = \frac{\|x_{t+1} - x_t\|}{\|u_t\|} \ , \tag{5}$$

A higher $\widehat{L}_t$ suggests greater state change per unit control, indicating higher local sensitivity.

3. **Threshold Update:** Based on the estimators, we update the threshold inversely to the estimators. Here we simply apply exponential descay.

$$\epsilon_t = \epsilon_0 \cdot \exp\left(-\alpha_L \widehat{L}_t\right) \cdot \exp\left(-\alpha_\delta \delta_t^o\right) . \tag{6}$$

---

**Algorithm 1** AdaReP: Adaptive Re-Planning Threshold Update (theoretical version)

**Require:** action $u_t$, prediction $y_{t+1}$, base threshold $\epsilon_t$
 1: Calculate prediction error $\delta_t^o$ using Equation (4)
 2: Estimate local dynamics sensitivity $\widehat{L}_t$ by Equation (5)
 3: Update the threshold $\epsilon_t$ by Equation (6)

---

We also provide theoretical regret analysis for this algorithm in Theorem C.7, which reveals how we can manipulate the additional regret by simply setting different values of $\epsilon_0$, $\alpha_L$, and $\alpha_\delta$, with no reliance on $L$ or $E$, which is not supported by non-adaptive methods like $\mathsf{MPC}_k^m$ or $\mathsf{MPC}_{k,\epsilon}$.

**Implementation Details**    Practical implementation of our algorithm involves several key details:

1. **Sliding Window:** The estimators calculated by Equation (4) and Equation (5) are quite noisy. To achieve more stable update, we apply sliding window to stabilize the adaptation against noisy single-step measurements. Our sensitive analysis in Section 4.3 demonstrate that sliding window method are crucial for practical implementations.

2. **State Distance Calculation:** The method for calculating distances, particularly the prediction error $\delta_{t+1}^o = \|x_{t+1} - y_{t+1}\|_2$ (Equation (4)), depends on the nature of the predictive world model:

   - For **state-based world models** that directly output predicted state vectors, we compute the L2-norm ($\|\cdot\|_2$) between the actual and predicted state vectors.

- For **vision-based world models** that output predicted images, calculating distances directly in pixel space is often ineffective. Instead, we first extract semantic features from both the actual observed image ($I(x_{t+1})$) and the predicted image ($I(y_{t+1})$) using a pre-trained feature extractor, such as DINO (Caron et al., 2021). Let $\phi(\cdot)$ denote this feature extraction function. The prediction error is then computed in the feature space:

$$\delta^o_{t+1} = \|\phi(I(x_{t+1})) - \phi(I(y_{t+1}))\|_2 \ . \tag{7}$$

3. **Hyper-parameters Tuning:** Our tuning is an upfront and efficient process as it builds upon the baseline. A practitioner can start with a reasonable fixed threshold $\epsilon$ and then simply tune $\alpha_L$ and $\alpha_\delta$ , which control the adaptation's sensitivity. Note that we do not need to tune the algorithm again if we switch the MPC planners (eg. MPPI,CEM).

Finally we remark that ADAREP is a training-free, plug-and-play module that can seamlessly be adapted to any world models and MPC planners. It can be adapted to **any** tasks that standard $\mathsf{MPC}^1_k$ can handle.

## 4 EXPERIMENTS

### 4.1 EXPERIMENT SETTINGS

Our experiments are designed to address the following key questions regarding our proposed adaptive replanning algorithm:

1. **Efficiency and Performance:** Does our proposed algorithm significantly accelerate the planning process compared to canonical MPC solvers while maintaining comparable or achieving even better task performance ? (Figure 2)

2. **Generalization:** Does the adaptive replanning mechanism generalize effectively across different predictive world models and diverse manipulation tasks within the simulated environment? (Figure 2)

3. **Real-World Applicability:** Can the benefits observed in simulation translate to challenging real-world robotic planning scenarios using learned world models? (Figure 3)

**Simulated Experiments.** We conduct simulated experiments using the VP2 benchmark (Tian et al., 2023b), a control-centric benchmark designed for evaluating video prediction models in manipulation tasks. This allows us to assess our algorithm's effectiveness and adaptability across various world models and tasks in a controlled setting. VP2 utilizes the RoboDesk simulation environment (Kannan et al., 2021) and provides pre-trained predictive world models relevant to this environment.

The RoboDesk environment features a Franka Emika Panda robot arm situated before a desk with various objects. We evaluate on the following 7 tasks defined within RoboDesk: pushing buttons (red, green, blue), opening a slide, opening a drawer, and pushing blocks (upright, flat) off the table.

We test our adaptive replanning approach with two distinct open-source video predictive world models provided by the VP2 benchmark: SVG (Villegas et al., 2019) and Struct-VRNN (Minderer et al., 2019). These models represent different architectural choices for video prediction.Here we compare our adaptive method $\mathsf{MPC}_{AR}$ against standard $\mathsf{MPC}^1_k$ and other non-adaptive method $\mathsf{MPC}^m_k$ and $\mathsf{MPC}_{k,\epsilon}$.

**Real-World Experiments.** To assess performance beyond simulation, we conduct real-world experiments with a Franka Emika Panda robotic arm, as detailed in Appendix E.2. we utilize state-based world models trained as described in Appendix E.3. We selected two challenging task categories representing different manipulation types prevalent in real-world scenarios:

1. **Open Door:** An articulation task requiring precise interaction with a hinged object. We evaluate on sub-tasks of opening the door to $90°$ and $180°$.

2. **Push T-Block:** A representative long-horizon rearrangement task. We define three sub-tasks: (i) translating the T-block to a target position, (ii) rotating the T-block to a target orientation, and (iii) a combined task of translating and then rotating the T-block.

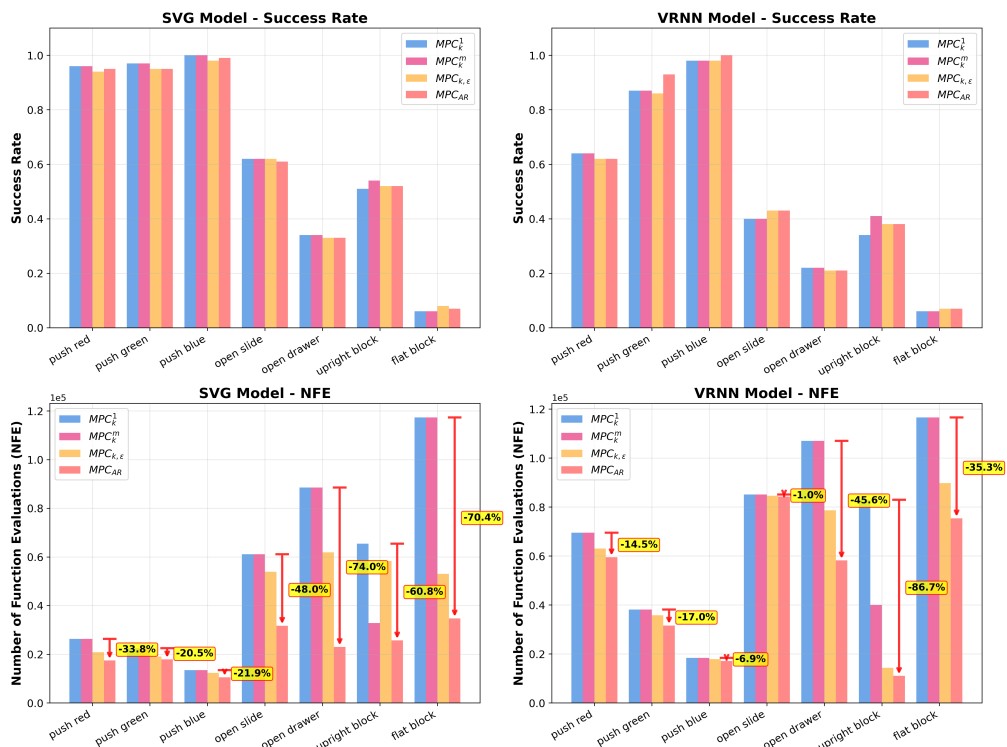

Figure 2: Results in Simulated Experiments. Here we tune the hyperparameter of the algorithms such that the performance are almost the same. It is demonstrated that our adaptive method enjoys most significant compuational savings.

## 4.2 MAIN RESULTS

**Results in Simulated Experiments**    For all the algorithms, we tune their hyperparameters such the success rates of all the accelerating methods drop no more than **0.02** compared with standard $MPC_k^1$. From Appendix F.1 we can see that both $MPC_k^m$ and $MPC_{k,\epsilon}$ cannot significantly accelerate compuation and guarantee performance simultaneously. $MPC_{AR}$, which adopts adaptive re-planning schedule, have much advantages as shown in Figure 2.

**Results in Real-World Experiments**    Visual demonstrations are detailed in Appendix F.2. Here we only provide quantitative results. As shown in Figure 3, our adaptive approach $MPC_{AR}$ demonstrates significant NFE reduction while maintaining or improving success rates across various sub-tasks. It has much better performance compared with that in the simulator because state-based planning is more explicit than vision-based planning.

## 4.3 ANAYLSIS AND DISCUSSIONS

**ADAREP achieves greater acceleration with accurate predictions.**    To investigate the impact of prediction accuracy on our adaptive algorithm, we conducted additional experiments using a "disturbed simulator" as the predictive world model which allowed us to directly control the magnitude of prediction errors introduced into the system. Specifically, we added varying levels of Gaussian noise to the true simulated state components (robot position/velocity, object position/velocity, and end-effector position) before feeding them to the planner. We evaluated three distinct levels of disturbance, with further details provided in Appendix E.6 and results are summarized in Figure 7.

**ADAREP achieves greater acceleration when system dynamics are smoother.**    The effective sensitivity of system dynamics can vary considerably, even within a single manipulation task. Consider the "open door" task: the dynamics are often highly sensitive when the end-effector interacts with

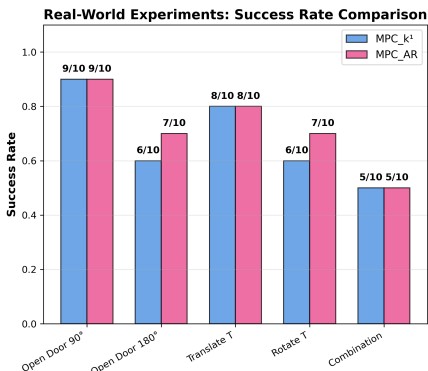 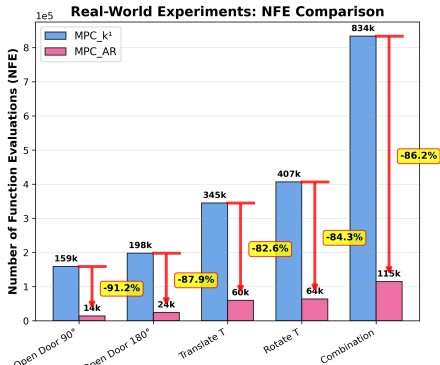

Figure 3: Results in Real-World Experiments. MPC$_{AR}$ can maintain strong performance and enjoy compuational savings simultaneously.

the door near its axis of rotation, as small end-effector movements can induce large angular changes. Figure 4 illustrates this concept. ADAREP can reduce re-planning frequency more aggressively, leading to greater computational savings.

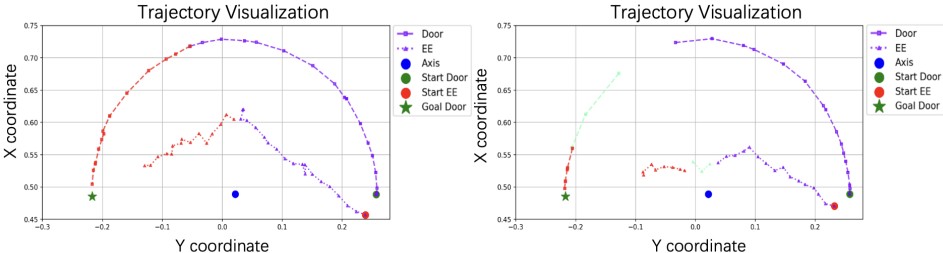

Figure 4: Illustration of varying dynamics sensitivity in the "open door" task. Left: Pushing the door far from its axis often results in more predictable, smoother changes in the door's state per unit of end-effector motion. Right: Pushing very close to the axis can lead to more abrupt or sensitive changes. Different colors represent controls from different plans.

**ADAREP maintains performance even in worst-case scenarios.** Real-world applications often present challenges such as large, unexpected prediction errors from the world model or highly sensitive, difficult-to-control system dynamics. In such adverse conditions, a key strength of MPC$_{AR}$ is its ability to adapt and prioritize task performance. By continuously monitoring metrics like prediction error (see Figure 5) and estimated local dynamics sensitivity (Figure 6), our algorithm automatically reduces its replanning threshold $\epsilon_t$. This leads to more frequent replanning, effectively causing MPC$_{AR}$ to behave more like standard MPC$_k^1$, thereby ensuring robustness and maintaining performance, albeit with reduced computational savings in these demanding situations.

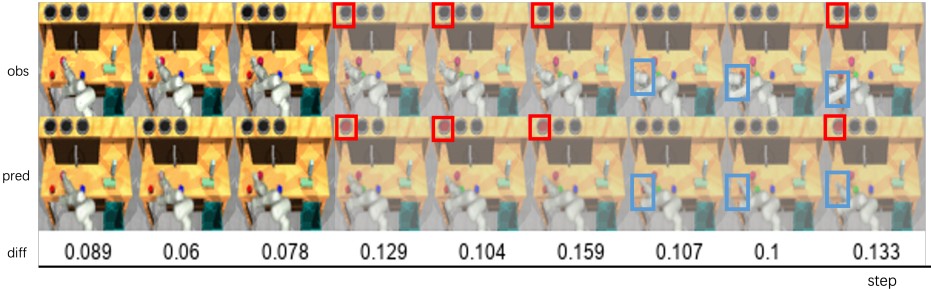

Figure 5: Visualization of prediction error monitoring. The first row shows observed images from the environment. The second row displays the corresponding images predicted by the world model. The third row quantifies the prediction error, likely computed in a feature space as Equation (4) using features from DINO Equation (7). Large discrepancies trigger more frequent replanning.

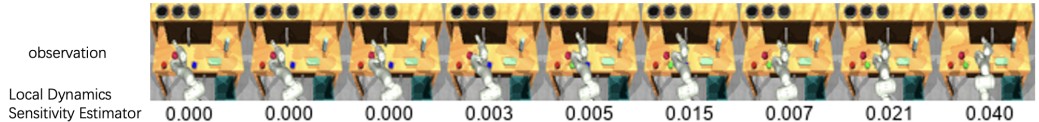

observation

Local Dynamics
Sensitivity Estimator

Figure 6: Visualization of local dynamics sensitivity estimation. The top row shows a sequence of observed frames. The bottom row displays the corresponding local dynamics sensitivity estimator $\widehat{L}_t$, computed using Equation (8). Higher values indicate more sensitive dynamics, prompting more frequent replanning.

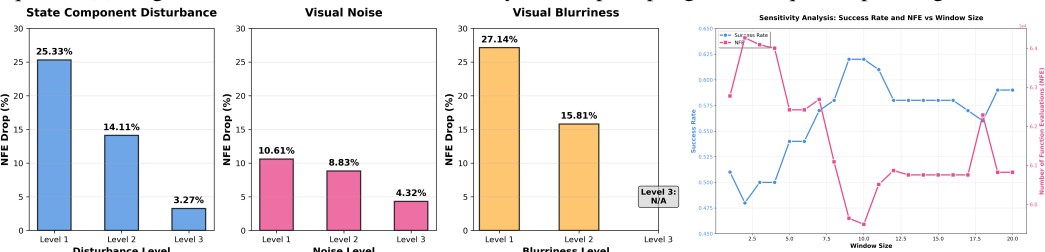

Figure 7: Impact of different levels of state component disturbance, visual noise, and visual blurriness on the NFE Drop of ADAREP (relative to standard MPC$_k^1$). Success rates were maintained at almost the same across all tests. Disturbance and corruption parameters are detailed in Appendix E.6 and Appendix E.7.

Figure 8: Sensitive Analysis on the window size. Small W (e.g., 1-5) results in poor performance while Large W (e.g., >15) makes performance plateaus.

**ADAREP performs better with more explicit state representations; robust visual features remain a challenge.** As observed in Section 4, ADAREP demonstrates significantly better effectiveness when coupled with state-based world models compared to vision-based ones. To investigate the impact of visual input quality on ADAREP when using vision-based world models, we simulated a scenario with a 'perfect' underlying state predictor but introduced various visual corruptions to the image outputs. This was achieved by adding different levels of Gaussian noise or applying Gaussian blur to the images generated by an otherwise accurate simulator (details in Appendix E.7). The results in Figure 7 indicate that even when the underlying state information provided to the planner is perfectly accurate (from the simulator), the performance of ADAREP (when relying on features extracted by DINO from these visually corrupted images) degrades significantly in terms of NFE reduction. This highlights the sensitivity to visual features and suggests that developing more robust visual feature extractors, or methods to better integrate them with adaptive MPC, is a crucial direction for future work.

**ADAREP is robust to the choice of window size.** We conduct additional experiments on the sensitive analysis of the window size, using the VRNN model. The result in Figure 8 reveals a clear and intuitive trade-off. Small W results in poor performance. We conjecture that the estimates are too unstable and noisy, leading to erratic replanning. Large W makes performance plateaus. The system becomes less responsive, averaging over too much history to react to recent changes. Crucially, the key takeaway is that ADAREP is robust to the choice of W. There is a wide range of values (W from 8 to 16) that yield strong and stable performance.

## 5 CONCLUSION

This research addressed the computational demands of Model Predictive Control (MPC) with learned world models by introducing ADAREP, an adaptive replanning strategy. Our theoretical analysis elucidated the interplay between replanning frequency, model prediction error, and local dynamics sensitivity, guiding the design of ADAREP which dynamically adjusts its planning effort based on online estimates. This training-free, plug-and-play approach demonstrated significant efficiency gains while preserving or enhancing task success rates. These findings underscore the potential of adaptive replanning for practical robotic control. Future work should focus on developing more robust visual features for vision-based models to broaden its applicability.

## 6 ETHICS STATEMENT AND REPRODUCIBILITY STATEMENT

**Ethics Statement**  We acknowledge and adhere to the ICLR Code of Ethics. This research focuses on improving computational efficiency in robotic control systems, which has broadly positive societal implications by making robotic automation more accessible and energy-efficient. Our work does not involve human subjects, and all experiments were conducted in controlled simulation environments and with standard robotic hardware. The datasets used (VP2 benchmark, RoboDesk) are publicly available research benchmarks with appropriate licenses. We do not foresee any direct harmful applications of our adaptive replanning methodology, as it is a general computational optimization technique. However, as with any robotics research, we acknowledge that improved robotic capabilities could potentially be misused in harmful applications, though this is far removed from our specific technical contributions. We have no conflicts of interest to declare, and this research was conducted with standard academic integrity practices.

**Reproducibility Statement**  We have made significant efforts to ensure the reproducibility of our work. Our adaptive replanning algorithm (ADAREP) is described in detail in Algorithm 1 with complete implementation details provided in Section 3. All experimental settings are specified in Section 4 and the appendix. The theoretical analysis is complete with full proofs provided in Appendix D. For the simulated experiments, we used the publicly available VP2 benchmark with standard evaluation protocols, and all experimental details are provided in Appendix E. The real-world experimental setup is thoroughly documented in Appendix E.2, including hardware specifications and data collection procedures. We plan to release our implementation code upon acceptance to facilitate reproduction of our results. The baseline methods are implemented using established algorithms from the literature with references provided.

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

## A  THE USE OF LARGE LANGUAGE MODELS (LLMS)

We acknowledge the use of Large Language Models (LLMs) in this work for the following specific tasks:

- **Writing assistance:** Grammar checking, sentence restructuring, and improving clarity of exposition.

- **Code implementation:** Implementing well-established baseline methods (CEM, MPPI) based on existing literature.

- **Visualization:** Generating and refining figures for data presentation.

**Importantly, all core contributions of this work were conceived and developed entirely by the authors:** the adaptive replanning idea, theoretical analysis and proofs, experimental design, novel algorithmic contributions, interpretation of results, and all scientific insights and conclusions. LLMs were not used for any creative, analytical, or decision-making aspects of the research.

## B  ALGORITHM PSEUDOCODES

---
**Algorithm 2** Model Predictive Control ($\text{MPC}_k^1$)

---
**Require:** Prediction horizon $k$, initial state $x(0)$, access to predictions $\xi_{\tau:\tau'|\tau}$, terminal cost function $F_T$. (Specify intermediate terminal costs $F_t$ for $k \leq t < T$ if needed for stability/performance).

1: **for** $t = 0, 1, \ldots, T - 1$ **do**
2: $\quad$ $t' \leftarrow \min\{t + k, T\}$
3: $\quad$ Observe current state $x_t$ and obtain predictions $\xi_{t:t'|t}$.
4: $\quad$ Define terminal cost for subproblem: $F_{term} = F_{t'}$ if $t' < T$ else $F_T$.
5: $\quad$ Define terminal parameter for subproblem: $\zeta_{term} = \xi_{t'|t}$.
6: $\quad$ Solve for $(y_{t:t'}, v_{t:t'-1}) = \psi_t^{t'}(x_t, \xi_{t:t'-1|t}, \zeta_{term}; F_{term})$.
7: $\quad$ Commit the first control action: $u_t := v_t$.

---

## C  THEORETICAL ANALYSIS

This section provides the detailed theoretical analysis underpinning our results, including assumptions, key technical lemmas, and regret bounds.

**Algorithm 3** Model Predictive Control with Fixed Replan Frequency ($\text{MPC}_k^m$)

---

**Require:** Prediction horizon $k$, replan frequency $m$ ($1 \leq m \leq k$), initial state $x(0)$, access to predictions $\xi_{\tau:\tau'|\tau}$, terminal cost function $F_T$. (Specify intermediate terminal costs $F_t$ for $k \leq t < T$ if needed).

  1: $t \leftarrow 0$
  2: **while** $t < T$ **do**
  3:     $t' \leftarrow \min\{t+k, T\}$
  4:     Observe current state $x_t$ and obtain predictions $\xi_{t:t'|t}$.
  5:     Define terminal cost for subproblem: $F_{term} = F_{t'}$ if $t' < T$ else $F_T$.
  6:     Define terminal parameter for subproblem: $\zeta_{term} = \xi_{t'|t}$.
  7:     Solve for $(y_{t:t'}, v_{t:t'-1}) = \psi_t^{t'}(x_t, \xi_{t:t'-1|t}, \zeta_{term}; F_{term})$.
  8:     Determine number of steps to commit: $m_{commit} = \min(m, T-t)$.
  9:     Commit the first $m_{commit}$ control actions: $u_\tau := v_\tau$ for $\tau = t, \ldots, t + m_{commit} - 1$.
 10:     $t \leftarrow t + m_{commit}$

---

**Algorithm 4** Model Predictive Control with Fixed Threshhold ($\text{MPC}_{k,\epsilon}$)

---

**Require:** Prediction horizon $k$, threshold $\epsilon$, initial state $x(0)$, access to predictions $\xi_{\tau:\tau'|\tau}$, terminal cost function $F_T$. (Specify intermediate terminal costs $F_t$ for $k \leq t < T$ if needed).

  1: $t \leftarrow 0$
  2: $t_{plan} \leftarrow 0$
  3: **while** $t < T$ **do**
  4:     **if** $t == t_{plan}$ **then**
  5:         $t' \leftarrow \min\{t+k, T\}$
  6:         Observe current state $x_t$ and obtain predictions $\xi_{t:t'|t}$.
  7:         Define terminal cost $F_{term} = F_{t'}$ if $t' < T$ else $F_T$.
  8:         Define terminal parameter $\zeta_{term} = \xi_{t'|t}$.
  9:         Solve for: $(y_{t:t'}, v_{t:t'-1}) = \psi_t^{t'}(x_t, \xi_{t:t'-1|t}, \zeta_{term}; F_{term})$.
 10:     $u_t := v_t$.
 11:     Execute action $u_t$ in environment.
 12:     Observe next state $x_{t+1}$.
 13:     **if** $\|x_{t+1} - y_{t+1}\| > \epsilon$ **then**
 14:         $t_{plan} \leftarrow t + 1$
 15:     **else**
 16:         **if** $t + 1 == t'$ **then**
 17:             $t_{plan} \leftarrow t + 1$
 18:     $t \leftarrow t + 1$

---

---

**Algorithm 5** Model Predictive Control with Adaptive Re-Planning (ADAREP) (MPC$_{AR}$)

---

**Require:** Prediction horizon $k$, threshold $\epsilon$, initial state $x(0)$, access to predictions $\xi_{\tau:\tau'|\tau}$, terminal cost function $F_T$. (Specify intermediate terminal costs $F_t$ for $k \leq t < T$ if needed).

1: $t \leftarrow 0$
2: $t_{plan} \leftarrow 0$
3: **while** $t < T$ **do**
4:     **if** $t == t_{plan}$ **then**
5:         $t' \leftarrow \min\{t + k, T\}$
6:         Observe current state $x_t$ and obtain predictions $\xi_{t:t'|t}$.
7:         Define terminal cost $F_{term} = F_{t'}$ if $t' < T$ else $F_T$.
8:         Define terminal parameter $\zeta_{term} = \xi_{t'|t}$.
9:         Solve for: $(y_{t:t'}, v_{t:t'-1}) = \psi_t^{t'}(x_t, \xi_{t:t'-1|t}, \zeta_{term}; F_{term})$.
10:     $u_t := v_t$.
11:     Execute action $u_t$ in environment.
12:     Observe next state $x_{t+1}$.
13:     Update threshold $\epsilon_{t+1}$ using Algorithm 1
14:     **if** $\|x_{t+1} - y_{t+1}\| > \epsilon_{t+1}$ **then**
15:         $t_{plan} \leftarrow t + 1$
16:     **else**
17:         **if** $t + 1 == t'$ **then**
18:             $t_{plan} \leftarrow t + 1$
19:     $t \leftarrow t + 1$

---

**Algorithm 6** ADAREP: Adaptive Re-Planning Threshold Update (practical version)

---

**Require:** Buffers $\mathcal{D}_\delta, \mathcal{D}_L$, action $u_t$, prediction $y_{t+1}$, base threshold $\epsilon_t$

1: Calculate prediction error $\delta_t^o$ using Eq. equation 4
2: Estimate local dynamics sensitivity $\widehat{L}_t$ by

$$\widehat{L}_t = \frac{\|x_{t+1} - x_t\|}{\|u_t\| + \varepsilon}, \tag{8}$$

3: Update buffer $\mathcal{D}_\delta \leftarrow \text{UPDATEBUF}(\mathcal{D}_\delta, \delta, W), \mathcal{D}_L \leftarrow \text{UPDATEBUF}(\mathcal{D}_L, \widehat{L}, W)$
4: Update the threshold $\epsilon_t \leftarrow \epsilon_0 \cdot \exp\left(-\alpha_\delta \text{mean}(\mathcal{D}_\delta)\right) \cdot \exp\left(-\alpha_L \text{mean}(\mathcal{D}_L)\right)$

---

Table 2: Summary of theoretical regret bounds and NFE. Here, $L = \max_{0 \le t < T} L_t$ and $L_* = \max_{0 \le t \le T} \max_{0 \le i \le m-1} \prod_{s=t}^{t+i} L_s$ characterize the sensitivity (Lipschitz constants) of the dynamics over single and multiple steps, respectively. $E$ represent cumulative prediction errors of the underlying model; they have slightly different forms but are conceptually similar for comparison purposes. Note that when $m = 1$ or $\epsilon = 0$, our results recover (Lin et al., 2022). And $S$ is the sample size of the planner at each step. $N_\epsilon, N_{\epsilon,\alpha} \le T$ is the total number of plans that decreases as $\epsilon, \alpha_L$ and $\alpha_\delta$ increases.

| Algorithm | Regret Bound | NFE |
|---|---|---|
| $\mathsf{MPC}_k^1$ | $O\left(\sqrt{L^2 \mathrm{cost}(\mathsf{OPT}) \cdot E} + L^2 E\right)$ | $TkS$ |
| $\mathsf{MPC}_k^m$ | $O\left(\sqrt{mL_*^2 \mathrm{cost}(\mathsf{OPT}) \cdot E} + mL_*^2 E\right)$ | $\lceil T/m \rceil kS$ |
| $\mathsf{MPC}_{k,\epsilon}$ | $O\left(\sqrt{L^2 \mathrm{cost}(\mathsf{OPT}) \cdot (E + \epsilon E + \epsilon^2 T)} + L^2(E + \epsilon E + \epsilon^2 T)\right)$ | $N_\epsilon kS$ |
| $\mathsf{MPC}_{AR}$ | $O\left(\sqrt{\mathrm{cost}(\mathsf{OPT})(L^2 E + \frac{\epsilon_0}{\alpha_L^2}(\epsilon_0 + \frac{1}{\alpha_\delta})T)} + L^2 E + \frac{\epsilon_0}{\alpha_L^2}(\epsilon_0 + \frac{1}{\alpha_\delta})T\right)$ | $N_{\epsilon,\alpha} kS$ |

## C.1 Assumptions and Notations

Our analysis largely follows the framework established by Lin et al. (2022). We impose the following standard assumptions throughout this section:

- **Stability of** OPT**:** The offline optimal trajectory $(x_{0:T}^*, u_{0:T-1}^*)$ is bounded. There exists a constant $D_{x^*} > 0$ such that $\|x_t^*\| \le D_{x^*}$ for all states $x_t^*$ on the optimal trajectory $(0 \le t \le T)$.

- **Lipschitz Dynamics:** The ground-truth dynamics function $g_t(\cdot, \cdot; \xi_t^*)$ is Lipschitz continuous with respect to both state and action. There exists a constant $L_t$ such that for any feasible states $x_t, x_t'$ and actions $u_t, u_t'$:

$$\|g_t(x_t, u_t; \xi_t^*) - g_t(x_t', u_t'; \xi_t^*)\| \le L_t(\|x_t - x_t'\| + \|u_t - u_t'\|). \tag{9}$$

- **Cost Function Regularity:** Every stage cost $f_t(\cdot, \cdot; \xi_t^*)$ and the terminal cost $F_T(\cdot; \xi_T^*)$ are non-negative, convex, and $\ell$-smooth with respect to $(x_t, u_t)$ and $x_T$, respectively, for some $\ell > 0$.

We note a slight strengthening compared to Lin et al. (2022), who only required the dynamics $g_t(\cdot, \cdot; \xi_t^*)$ to be Lipschitz continuous with respect to the action $u_t$. Our stronger assumption (Lipschitz continuity w.r.t. both state $x_t$ and action $u_t$) is utilized specifically in the analysis of $\mathsf{MPC}_k^m$ (Algorithm 3). The analysis for $\mathsf{MPC}_{k,\epsilon}$ (Algorithm 4) does not require this modification and holds under the weaker assumption.

When the context is clear, we use the shorthand $g_t(\cdot, \cdot) := g_t(\cdot, \cdot; \xi_t^*)$, $f_t(\cdot, \cdot) := f_t(\cdot, \cdot; \xi_t^*)$, and $F_T(\cdot) := F_T(\cdot; \xi_T^*)$ to simplify notation.

## C.2 Perturbation Analysis

Our analysis relies heavily on perturbation bounds for the Finite-Time Optimal Control Problem (FTOCP, see Definition 2.1), which characterize how the optimal solution changes in response to perturbations in parameters or initial states. Prior works (Shin et al., 2020; Lin et al., 2021; Shin & Zavala, 2021; Xu & Anitescu, 2019; Na & Anitescu, 2022) have established such bounds, often locally, for various FTOCP instances. We adopt the generalized forms presented by Lin et al. (2022):

(a) *Parameter Perturbations (fixed initial state $z$):* Let $\psi_{t_1}^{t_2}(z, \xi; F)_v$ denote the optimal control sequence from the FTOCP solution. Then,

$$\left\| \psi_{t_1}^{t_2}(z, \xi_{t_1:t_2}; F)_{v_t} - \psi_{t_1}^{t_2}(z, \xi_{t_1:t_2}'; F)_{v_t} \right\| \le \left( \sum_{s=t_1}^{t_2} q_1(s - t_1)\delta_s \right) \|z\| + \sum_{s=t_1}^{t_2} q_2(s - t_1)\delta_s, \tag{10}$$

where $\delta_s := \|\xi_s - \xi_s'\|$ for $s \in [t_1, t_2]$. The scalar functions $q_1, q_2$ represent sensitivity decay and satisfy $\lim_{t \to \infty} q_i(t) = 0$ and $\sum_{t=0}^{\infty} q_i(t) \le C_i$ for constants $C_i \ge 1$, $i = 1, 2$.

(b) *Initial State Perturbation (fixed parameters $\xi$):* Let $\psi_{t_1}^{t_2}(z, \xi; F)_{y_t/v_t}$ denote the state or control component at time $t$ of the optimal solution. Then,

$$\left\| \psi_{t_1}^{t_2}(z, \xi_{t_1:t_2}; F)_{y_t/v_t} - \psi_{t_1}^{t_2}(z', \xi_{t_1:t_2}; F)_{y_t/v_t} \right\| \le q_3(t-t_1) \left\| z - z' \right\|, \text{ for } t \in [t_1, t_2], \quad (11)$$

where the sensitivity decay function $q_3$ satisfies $\sum_{t=0}^{\infty} q_3(t) \le C_3$ for some constant $C_3 \ge 1$.

Intuitively, bound equation 10 suggests that errors in parameter predictions further in the future have a diminishing impact on the current optimal action. Bound equation 11 implies a form of stability: the effect of an initial state perturbation decays over time within the planned trajectory.

While these perturbation bounds are powerful, proving they hold globally can be challenging; often, they are established locally around a nominal trajectory. For a practical predictive control system designed to track an optimal trajectory, it is reasonable to expect the executed trajectory to remain relatively close to the (unknown) optimal one. Building on this, we adopt the following property, similar to Lin et al. (2022), which posits that these bounds hold within a certain region around the optimal trajectory OPT. Let $\mathcal{B}(x, R)$ denote the closed ball of radius $R$ centered at $x$.

**Property C.1.** *There exists a constant $R_1 > 0$ such that the perturbation bounds equation 10 and equation 11 hold under the following specifications, assuming the underlying parameter sets $\Xi_t$ contain the relevant parameters:*

- *Bound equation 10 holds for $t_1 = t, t_2 = t + k$ (where $t < T - k$) with terminal function $F = \mathbb{I}$ (identity), initial state $z \in \mathcal{B}(x_t^*, R_1)$, parameters $\xi'_{t:t+k-1} = \xi^*_{t:t+k-1}$ (ground truth), and $\xi_{t:t+k}$ being any valid parameters within the family.*

- *Bound equation 10 holds for $t_1 = t, t_2 = T$ (where $t \ge T - k$) with terminal function $F = F_T$, initial state $z \in \mathcal{B}(x_t^*, R_1)$, parameters $\xi'_{t:T} = \xi^*_{t:T}$, and $\xi_{t:T}$ being any valid parameters.*

- *Bound equation 11 holds for any $t_1, t_2$, any initial states $z, z' \in \mathcal{B}(x_{t_1}^*, R_1)$, and ground-truth parameters $\xi_{t_1:t_2} = \xi^*_{t_1:t_2}$.*

We quantify the quality of the parameter predictions $\xi_{t+\tau|t}$ (prediction of $\xi^*_{t+\tau}$ made at time $t$) available to the online controller.

**Definition C.1** (Prediction Error). *The prediction error at time $t$ for lead time $\tau \ge 0$ is $\rho_{t,\tau} :=$ $\left\| \xi_{t+\tau|t} - \xi^*_{t+\tau} \right\|$.*

A key challenge in analyzing online algorithms via regret is the state mismatch: the online algorithm's state $x_t$ generally differs from the offline optimal state $x_t^*$. Directly comparing the online action $u_t$ to the offline optimal action $u_t^*$ is therefore insufficient. Inspired by techniques in reinforcement learning (Lin et al., 2021), Lin et al. (2022) utilized a per-step error comparing the online action $u_t$ to the optimal action $u_{t|t}^*$ from the *current* state $x_t$. For analyzing $\mathsf{MPC}_k^m$, where actions are based on plans made at earlier times $t'$, we introduce a conditional variant.

**Definition C.2** (Per-Step Error). *The per-step error $e_t$ incurred by a predictive online controller* ALG *at time step $t$ is defined as the distance between its actual action $u_t$ and the clairvoyant optimal action, i.e.,*

$$e_t := \left\| u_t - u_{t|t}^* \right\| = \left\| u_t - \psi_t^T(x_t, \xi^*_{t:T}; F_T)_{v_t} \right\|, \text{ where } u_t = \mathsf{ALG}(x_t, \xi_{t:t+k|t})_{v_t}.$$

*The clairvoyant optimal trajectory starting from $x_t$ is defined as $x^*_{t:T|t} := \psi_t^T(x_t, \xi^*_{t:T}; F_T)_{y_{t:T}}$.*

**Definition C.3** (Conditional Per-Step Error). *The conditional per-step error $e_t$ incurred by a predictive online controller* ALG *at time step $t$ given time step $t'$ is defined as the distance between its actual action $u_t$ and the clairvoyant optimal action given $t'$*

$$e_{t|t'} := \left\| u_t - u_{t|t'}^* \right\| = \left\| u_t - \psi_{t'}^T(x_{t'}, \xi^*_{t':T}; F_T)_{v_t} \right\|, \text{ where } u_t = \mathsf{ALG}(x_{t'}, \xi_{t':t'+k|t'})_{v_t}.$$

*The clairvoyant optimal trajectory starting from $x_t$ is defined as $x^*_{t:T|t} := \psi_t^T(x_t, \xi^*_{t:T}; F_T)_{y_{t:T}}$.*

What's described above are adequate to analyze the regret of $\mathsf{MPC}_k^m$ (Algorithm 3) and $\mathsf{MPC}_{k,\epsilon}$ (Algorithm 4). However, when it comes to the analysis of $\mathsf{MPC}_{AR}$ (Algorithm 5), additional assumptions have to be made.

The core idea of ADAREP is to monitor and estimate local dynamic sensitivity and prediction error based on observation. Previous assumptions only characterize how performance or deviation can be upper bounder by local dynamic sensitivity and prediction error, but do not reveal how local dynamic sensitivity and prediction error can be estimated by ground-truth observation. In the light of this, we introduce the following properties, which gives the inequalities on the opposite direction of eq. (9) and Equation (10).

- **Lipschitz Dynamics:** For all $\|u_t\| > 0$, there exist $0 < \lambda < 0$ such that

$$\lambda L_t \|u_t\| \leq \|g_t(x_t, u_t; \xi_t^*) - x_t\| \leq L_t \|u_t\| \ .$$

- **Parameter Perturbations:** There exist $0 < \mu < 0$ such that

$$\left\| \psi_{t_1}^{t_2}\left(z, \xi_{t_1:t_2}; F\right)_{y_t} - \psi_{t_1}^{t_2}\left(z, \xi_{t_1:t_2}'; F\right)_{y_t} \right\| \geq \mu \left( \sum_{s=t_1}^{t_2} q_1(s - t_1)\delta_s \right) \|z\| + \mu \sum_{s=t_1}^{t_2} q_2(s-t_1)\delta_s, \tag{12}$$

Now we are ready to provide the regret analysis for all the algorithms.

### C.3 REGRET ANALYSIS FOR $\mathsf{MPC}_k^m$

We first bound the conditional per-step error for the $\mathsf{MPC}_k^m$ algorithm.

**Lemma C.1** (Conditional Per-Step Error Bound). *Assume Property C.1 holds. Let $t' = mn \leq t < m(n + 1)$. Assume the state at the last replanning time satisfies $x_{t'} \in \mathcal{B}(x_{t'}^*, R_1)$. Further assume the (potentially hypothetical) terminal cost used within the FTOCP solved at time $t'$ implies a target terminal state $\bar{y}(\xi_{t'+k|t'}) \in \mathcal{B}(x_{t'+k}^*, R_2)$ for some constant $R_2 \geq R_1 > 0$. Then, the conditional per-step error $e_{t|t'}$ of $\mathsf{MPC}_k^m$ is bounded by:*

$$e_{t|t'} \leq \sum_{\tau=0}^{k} \left( (R_1 + D_{x^*}) \cdot q_1(\tau) + q_2(\tau) \right) \rho_{t',\tau} + 2R_2 \left( (R_1 + D_{x^*}) \cdot q_1(k) + q_2(k) \right) \ . \tag{13}$$

Next, we relate the deviation from the clairvoyant optimal trajectory (starting from $x_{t'}$) to these conditional errors.

**Lemma C.2** (State Deviation Bound). *Let $t' = mn \leq t < m(n+1)$ and let $x_{\tau|t'}^*$ denote the state at time $\tau$ on the clairvoyant optimal trajectory starting from $x_{t'}$ at time $t'$. Under the Lipschitz dynamics assumption, we have:*

$$\left\| x_t - x_{t|t'}^* \right\| \leq \sum_{\tau=t'}^{t-1} e_{\tau|t'} \left( \prod_{s=\tau+1}^{t-1} L_s \right) \ . \tag{14}$$

The following lemma connects the cumulative conditional per-step errors to the dynamic regret, analogous to Lemma 3.2 in Lin et al. (2022).

**Lemma C.3** (Regret Bound via Conditional Errors). *Assume $T = mN$ for integer $N$. Let $L_* = \max_{0 \leq t \leq T-k} \prod_{s=t}^{t+k-1} L_s$. Under the assumptions on cost function regularity and the applicability of perturbation bound equation 11 (Property C.1), the dynamic regret of $\mathsf{MPC}_k^m$ is upper bounded by:*

$$\mathrm{cost}(\mathsf{MPC}_k^m) - \mathrm{cost}(\mathsf{OPT})$$

$$\leq \sqrt{\left( \frac{\ell}{2} \cdot 2mL_*^2 C_3^2 \right) \cdot \mathrm{cost}(\mathsf{OPT}) \cdot \left( \sum_{i=0}^{N-1} \sum_{\tau=mi}^{m(i+1)-1} e_{\tau|mi}^2 \right)}$$

$$+ \frac{\ell}{2} \cdot 2mL_*^2 C_3^2 \left( \sum_{i=0}^{N-1} \sum_{\tau=mi}^{m(i+1)-1} e_{\tau|mi}^2 \right) \ . \tag{15}$$

Combining these results yields the main regret theorem for $\mathsf{MPC}_k^m$.

**Theorem C.4** (Regret Bound for $\mathsf{MPC}_k^m$). *Let Property C.1 hold. Suppose the terminal cost $F_{t+k}$ of $\mathsf{MPC}_k^m$ is set to be the indicator function of some state $\bar{y}(\xi_{t+k|t})$ that satisfies $\bar{y}(\xi_{t+k|t}) \in \mathcal{B}(x_{t+k}^*, R_2)$ for all time steps $t < T - k$. Further, suppose the prediction errors $\rho_{t,\tau}$ are sufficiently small and the prediction horizon $k$ is sufficiently large, such that*

$$\sum_{\tau=0}^{k} \left((R_1 + D_{x^*}) \cdot q_1(\tau) + q_2(\tau)\right) \rho_{mn,\tau} + 2R_2 \left((R_1 + D_{x^*}) \cdot q_1(k) + q_2(k)\right) \leq \frac{R_1}{mC_3 L_*} .$$

*Then, the trajectory of $\mathsf{MPC}_k^m$ will remain close to $\mathsf{OPT}$, i.e. $x_t \in \mathcal{B}(x_t^*, R_1)$ for all time steps $t$, and the dynamic regret of $\mathsf{MPC}_k^m$ is upper bounded by*

$$\text{cost}(\mathsf{MPC}_k^m) - \text{cost}(\mathsf{OPT}) = O\left(\sqrt{mL_*^2\text{cost}(\mathsf{OPT}) \cdot E_1} + mL_*^2 E_1\right), \tag{16}$$

*where $E_1 = O\left(\sum_{\tau=0}^{k-1} (R_0 \cdot q_1(\tau) + q_2(\tau)) \sum_{n=0}^{N} m\rho_{mn,\tau}^2 + \left(q_1(k)^2 + q_2(k)^2\right) T\right)$.*

## C.4 REGRET ANALYSIS FOR $\mathsf{MPC}_{k,\epsilon}$.

We first bound the per-step error of $\mathsf{MPC}_{k,\epsilon}$.

**Lemma C.5.** *Let Property C.1 hold. Suppose the current state $x_t$ satisfies $x_t \in \mathcal{B}(x_t^*, R_1)$ and the terminal cost $F_{t+k}$ of $\mathsf{MPC}_{k,\epsilon}$ is set to be the indicator function of some state $\bar{y}(\xi_{t+k|t})$ that satisfies $\bar{y}(\xi_{t+k|t}) \in \mathcal{B}(x_{t+k}^*, R_2)$ for $t < T - k$. Further, suppose the last planned time step is $t'$. Then, the per-step error of $\mathsf{MPC}_{k,\epsilon}$ and $\mathsf{MPC}_{AR}$ is bounded by*

$$e_t \leq q_3(0)\epsilon + \sum_{\tau=0}^{k} \left((R_1 + D_{x^*}) \cdot q_1(\tau) + q_2(\tau)\right) \rho_{t',\tau} + 2R_2 \left((R_1 + D_{x^*}) \cdot q_1(k) + q_2(k)\right) . \tag{17}$$

This yields the final regret bonud for $\mathsf{MPC}_{k,\epsilon}$.

**Theorem C.6** (Regret Bound for $\mathsf{MPC}_{k,\epsilon}$). *Let Property C.1 hold. Suppose the terminal cost $F_{t+k}$ of $\mathsf{MPC}_k^m$ is set to be the indicator function of some state $\bar{y}(\xi_{t+k|t})$ that satisfies $\bar{y}(\xi_{t+k|t}) \in \mathcal{B}(x_{t+k}^*, R_1)$ for all time steps $t < T - k$. Let $L = \max_{0 \leq t \leq T} L_t$ Further, suppose the prediction errors $\rho_{t,\tau}$ and threshold $\epsilon$ are sufficiently small and the prediction horizon $k$ is sufficiently large, such that*

$$q_3(0)\epsilon + \sum_{\tau=0}^{k} \left((R_1 + D_{x^*}) \cdot q_1(\tau) + q_2(\tau)\right) \rho_{t,\tau} + 2R_2 \left((R_1 + D_{x^*}) \cdot q_1(k) + q_2(k)\right) \leq \frac{R_1}{C_3 L} .$$

*Then, the trajectory of $\mathsf{MPC}_{k,\epsilon}$ will remain close to $\mathsf{OPT}$, i.e. $x_t \in \mathcal{B}(x_t^*, R_1)$ for all time steps $t$, and the dynamic regret of $\mathsf{MPC}_{k,\epsilon}$ is upper bounded by*

$$\text{cost}(\mathsf{MPC}_{k,\epsilon}) - \text{cost}(\mathsf{OPT}) = O\left(\sqrt{L^2\text{cost}(\mathsf{OPT}) \cdot (E_2 + \epsilon E_2 + \epsilon^2 T)} + mL_*^2(E_2 + \epsilon E_2 + \epsilon^2 T)\right),$$
$$\tag{18}$$

*where $E_2 = O\left(\sum_{\tau=0}^{k-1} (R_0 \cdot q_1(\tau) + q_2(\tau)) \sum_{t=0}^{T} \rho_{p(t),\tau}^2 + \left(q_1(k)^2 + q_2(k)^2\right) T\right)$, $p(t)$ denotes the last time step that plans before $t$.*

## C.5 REGRET ANALYSIS FOR $\mathsf{MPC}_{AR}$.

**Theorem C.7** (Regret Bound for $\mathsf{MPC}_{AR}$). *Let Property C.1 hold. Suppose the terminal cost $F_{t+k}$ of $\mathsf{MPC}_k^m$ is set to be the indicator function of some state $\bar{y}(\xi_{t+k|t})$ that satisfies $\bar{y}(\xi_{t+k|t}) \in \mathcal{B}(x_{t+k}^*, R_1)$ for all time steps $t < T - k$. Let $L = \max_{0 \leq t \leq T} L_t$ Further, suppose the prediction errors $\rho_{t,\tau}$ and threshold $\epsilon$ are sufficiently small and the prediction horizon $k$ is sufficiently large, such that*

$$q_3(0)\epsilon + \sum_{\tau=0}^{k} \left((R_1 + D_{x^*}) \cdot q_1(\tau) + q_2(\tau)\right) \rho_{t,\tau} + 2R_2 \left((R_1 + D_{x^*}) \cdot q_1(k) + q_2(k)\right) \leq \frac{R_1}{C_3 L} .$$

*Then, the trajectory of $\mathsf{MPC}_{k,\epsilon}$ will remain close to $\mathsf{OPT}$, i.e. $x_t \in \mathcal{B}(x_t^*, R_1)$ for all time steps $t$, and the dynamic regret of $\mathsf{MPC}_{k,\epsilon}$ is upper bounded by*

$$\text{cost}(\text{MPC}_{k,\epsilon}) - \text{cost}(\text{OPT}) \tag{19}$$

$$= O\left(\min\left\{ Regret(\text{MPC}_{k,\epsilon_0}), \sqrt{\text{cost}(\text{OPT})(L^2 E + \frac{\epsilon_0}{\alpha_L^2}(\epsilon_0 + \frac{1}{\alpha_\delta})T)} + L^2 E + \frac{\epsilon_0}{\alpha_L^2}(\epsilon_0 + \frac{1}{\alpha_\delta})T \right\}\right),$$

where $E_2 = O\left(\sum_{\tau=0}^{k-1}(R_0 \cdot q_1(\tau) + q_2(\tau))\sum_{t=0}^{T}\rho_{p(t),\tau}^2 + (q_1(k)^2 + q_2(k)^2)T\right)$, $p(t)$ denotes the last time step that plans before $t$.

## D  Technical Proof

### D.1  Proof for $\text{MPC}_k^m$

*Proof of Lemma C.1.* Lemma 20 is a straight-forward implication of perturbation bound equation 10 To see this, for $t' = mn \le t < m(n+1)$, note that the per-step error $e_t$ can be bounded by

$$e_{t|t'} = \left\| \psi_{t'}^{t'+k}(x_{t'}, \xi_{t':t'+k-1|t'}, \bar{y}(\xi_{t'+k|t'}); \mathbb{I})_{v_t} - \psi_t^T(x_t, \xi_{t:T}^*; F_T)_{v_t} \right\| \tag{20a}$$

$$= \left\| \psi_{t'}^{t'+k}(x_{t'}, \xi_{t':t'+k-1|t'}, \bar{y}(\xi_{t'+k|t'}); \mathbb{I})_{v_t} - \psi_{t'}^{t'+k}(x_{t'}, \xi_{t':t'+k-1}^*, x_{t'+k|t'}^*; \mathbb{I})_{v_t} \right\| \tag{20b}$$

$$\le \sum_{\tau=0}^{k-1}\left(\|x_{t'}\| \cdot q_1(\tau) + q_2(\tau)\right)\rho_{t',\tau} + \left(\|x_{t'}\| \cdot q_1(k) + q_2(k)\right)\left\| \bar{y}(\xi_{t'+k|t'}) - x_{t'+k|t'}^* \right\|. \tag{20c}$$

Here, we apply the principle of optimality to conclude that the optimal trajectory from $x_{t'}$ to $x_{t'+k|t'}^*$ (i.e., $\psi_{t'}^{t'+k}(x_{t'}, \xi_{t':t'+k-1}^*, x_{t'+k|t'}^*; \mathbb{I})$ in equation 20b) is a sub-trajectory of the clairvoyant optimal trajectory from $x_t$ (i.e., $\psi_t^T(x_t, \xi_{t:T}^*; F_T)$ in equation 20a), and equation 20c is obtained by directly applying perturbation bound equation 10. Note that $\|x_{t'}\| \le R_1 + D_{x^*}$, and that both $\bar{y}(\xi_{t'+k|t'})$ and $x_{t'+k|t'}^*$ are in $\mathcal{B}(x_{t+k}^*; R)$ by assumption and by perturbation bound equation 11 specified in Property C.1, we conclude that equation 13 hold for $t < T - k$. The case $t \ge T - k$ can be shown similarly. $\square$

*Proof of Lemma C.2.* We use mathematical induction to show how state deviations accumulate when executing multiple actions from a single plan. The key insight is that each action error compounds through the system dynamics.

We prove by induction for the first part. When $t = t' + 1$ (i.e., the first step after planning):

$$\left\| x_{t'+1} - x_{t'+1|t'}^* \right\| = \left\| g_{t'}(x_{t'}, u_{t'}) - g_{t'}(x_{t'}, u_{t'|t'}^*) \right\|$$

$$\le L_{t'}\left\| u_{t'} - u_{t'|t'}^* \right\|$$

$$\le L_{t'}e_{t'|t'}.$$

Now suppose eq. (14) holds for $t - 1$ (i.e., $\left\| x_t - x_{t-1|t'}^* \right\| \le \sum_{\tau=t'}^{t-2} e_{\tau|t'}\prod_{s=\tau}^{t-2} L_s$). This means we assume the state deviation bound holds up to time $t - 1$. We now show it holds for time $t$:

$$\left\| x_t - x_{t|t'}^* \right\| = \left\| g_{t-1}(x_{t-1}, u_{t-1}) - g(x_{t-1|t'}^*, u_{t-1|t'}^*) \right\|$$

$$\le L_{t-1}(\left\| u_{t-1} - u_{t-1|t'}^* \right\| + (\left\| x_{t-1} - x_{t-1|t'}^* \right\|)$$

$$\le L_{t-1}e_{t-1|t'} + L_{t-1}\sum_{\tau=t'}^{t-2} e_{\tau|t'}\prod_{s=\tau}^{t-2} L_s$$

$$= \sum_{\tau=t'}^{t-1} e_{\tau|t'}\prod_{s=\tau}^{t-1} L_s.$$

$\square$

*Proof of Lemma C.3.* First we bound $\|x_t - x_t^*\|$ and $\|u_t - u_t^*\|$

$$\|x_t - x_t^*\| = \left\|x_t - x_{t|mn}^*\right\| + \left\|x_{t|mn}^* - x_t^*\right\|$$

$$\leq \left\|x_t - x_{t|mn}^*\right\| + \sum_{i=0}^{n-1} \left\|x_{t|m(i+1)}^* - x_{t|mi}^*\right\|$$

$$\leq \left\|x_t - x_{t|mn}^*\right\| + \sum_{i=0}^{n-1} q_3(t-(i+1)m) \left\|x_{m(i+1)} - x_{m(i+1)|mi}^*\right\|$$

$$\leq \sum_{\tau=mn}^{t-1} e_{\tau|mn} \prod_{s=\tau}^{t-1} L_s + \sum_{i=0}^{n-1} q_3(t-(i+1)m) \sum_{\tau=mi}^{m(i+1)-1} e_{\tau|mi} \prod_{s=\tau}^{m(i+1)-1} L_s \ .$$

$$\|u_t - u_t^*\| = \left\|u_t - u_{t|mn}^*\right\| + \left\|u_{t|mn}^* - u_t^*\right\|$$

$$\leq \left\|u_t - u_{t|mn}^*\right\| + \sum_{i=0}^{n-1} \left\|u_{t|m(i+1)}^* - u_{t|mi}^*\right\|$$

$$\leq \left\|u_t - u_{t|mn}^*\right\| + \sum_{i=0}^{n-1} q_3(t-(i+1)m) \left\|x_{m(i+1)} - x_{m(i+1)|mi}^*\right\|$$

$$\leq e_{t|mn} + \sum_{i=0}^{n-1} q_3(t-(i+1)m) \sum_{\tau=mi}^{m(i+1)-1} e_{\tau|mi} \prod_{s=\tau}^{m(i+1)-1} L_s \ .$$

Without loss of generality, we define $q_3(k) = 0$ for $k < 0$. For simplicity of notation, we denote $L_* = \max_{0 \leq t \leq T} \max_{0 \leq i \leq m-1} \prod_{s=t}^{t+i} L_s$, which captures the maximum compounding effect of Lipschitz constants over $m$ steps. So we arrive at

$$\|x_t - x_t^*\|, \|u_t - u_t^*\| \leq L_* \sum_{i=0}^{n} q_3(t-(i+1)m) \sum_{\tau=mi}^{m(i+1)-1} e_{\tau|mi} \ . \tag{21}$$

To bound the squared deviations (which will be needed for the cost analysis), we use the Cauchy-Schwarz inequality:

$$\|x_t - x_t^*\|^2 \leq L_*^2 \left( \sum_{i=0}^{n} q_3(t-(i+1)m) \sum_{\tau=mi}^{m(i+1)-1} e_{\tau|mi} \right)^2$$

$$\leq L_*^2 \sum_{i=0}^{n} q_3(t-(i+1)m)) \left( \sum_{i=0}^{n} q_3(t-(i+1)m) \left( \sum_{\tau=mi}^{m(i+1)-1} e_{\tau|mi} \right)^2 \right)$$

$$\leq m L_*^2 C_3 \left( \sum_{i=0}^{n} q_3(t-(i+1)m) \sum_{\tau=mi}^{m(i+1)-1} e_{\tau|mi}^2 \right) \ .$$

This bound also holds for $\|u_t - u_t^*\|^2$, so we have

$$\|x_t - x_t^*\|^2 + \|u_t - u_t^*\|^2 \leq 2m L_*^2 C_3 \left( \sum_{i=0}^{n} q_3(t-(i+1)m) \sum_{\tau=mi}^{m(i+1)-1} e_{\tau|mi}^2 \right) \ .$$

Without loss of generality, we assue $T = Nm$, we have

$$\sum_{t=1}^{T} \|x_t - x_t^*\|^2 + \sum_{t=1}^{T} \|u_t - u_t^*\|^2 \leq 2mL_*^2C_3 \left( \sum_{n=0}^{N-1} \sum_{t=nm}^{(n+1)m-1} \sum_{i=0}^{n} q_3(t-(i+1)m) \sum_{\tau=mi}^{m(i+1)-1} e_{\tau|mi}^2 \right)$$

$$\leq 2mL_*^2C_3 \left( \sum_{i=0}^{N-1} \sum_{\tau=mi}^{m(i+1)-1} e_{\tau|mi}^2 \sum_{n=i}^{N-1} \sum_{t=nm}^{(n+1)m-1} q_3(t-(i+1)m) \right)$$

$$\leq 2mL_*^2C_3^2 \left( \sum_{i=0}^{N-1} \sum_{\tau=mi}^{m(i+1)-1} e_{\tau|mi}^2 \right) \tag{22}$$

Since the cost function $f_t(\cdot, \cdot; \xi_t^*)$ and $F_T(\cdot; \xi_T^*)$ are nonnegative, convex, and $\ell$-smooth in their inputs, by Lemma F.2 in Lin et al. (2021), we see that the following inequality holds for arbitrary $\eta > 0$:

$$\text{cost}(\mathsf{ALG}) - \text{cost}(\mathsf{OPT})$$

$$\leq \left( \sum_{t=0}^{T-1} f_t(x_t, u_t; \xi_t^*) + F_T(x_T; \xi_T^*) \right) - \left( \sum_{t=0}^{T-1} f_t(x_t^*, u_t^*; \xi_t^*) + F_T(x_T^*; \xi_T^*) \right)$$

$$\leq \eta \left( \sum_{t=0}^{T-1} f_t(x_t^*, u_t^*; \xi_t^*) + F_T(x_T^*; \xi_T^*) \right)$$

$$+ \frac{\ell}{2} \left( 1 + \frac{1}{\eta} \right) \left( \sum_{t=1}^{T} \|x_t - x_t^*\|^2 + \sum_{t=0}^{T-1} \|u_t - u_t^*\|^2 \right) \tag{23a}$$

$$\leq \eta \cdot \text{cost}(\mathsf{OPT}) + \left( 1 + \frac{1}{\eta} \right) \cdot \frac{\ell}{2} \cdot 2mL_*^2C_3^2 \left( \sum_{i=0}^{N-1} \sum_{\tau=mi}^{m(i+1)-1} e_{\tau|mi}^2 \right) \tag{23b}$$

$$= \eta \cdot \text{cost}(\mathsf{OPT}) + \frac{1}{\eta} \cdot \frac{\ell}{2} \cdot 2mL_*^2C_3^2 \left( \sum_{i=0}^{N-1} \sum_{\tau=mi}^{m(i+1)-1} e_{\tau|mi}^2 \right)$$

$$+ \frac{\ell}{2} \cdot 2mL_*^2C_3^2 \left( \sum_{i=0}^{N-1} \sum_{\tau=mi}^{m(i+1)-1} e_{\tau|mi}^2 \right), \tag{23c}$$

where we apply Lemma F.2 in Lin et al. (2021) in equation 23a, and we use equation 22 in equation 23b. Setting the tunable weight $\eta$ in equation 23c to be

$$\eta = \left( \frac{\frac{\ell}{2} \cdot 2mL_*^2C_3^2 \left( \sum_{i=0}^{N-1} \sum_{\tau=mi}^{m(i+1)-1} e_{\tau|mi}^2 \right)}{\text{cost}(\mathsf{OPT})} \right)^{\frac{1}{2}}$$

gives that

$$\text{cost}(\mathsf{ALG}) - \text{cost}(\mathsf{OPT})$$

$$\leq \sqrt{\left( \frac{\ell}{2} \cdot 2mL_*^2C_3^2 \right) \cdot \text{cost}(\mathsf{OPT}) \cdot \left( \sum_{i=0}^{N-1} \sum_{\tau=mi}^{m(i+1)-1} e_{\tau|mi}^2 \right)}$$

$$+ \frac{\ell}{2} \cdot 2mL_*^2C_3^2 \left( \sum_{i=0}^{N-1} \sum_{\tau=mi}^{m(i+1)-1} e_{\tau|mi}^2 \right). \tag{24}$$

This finishes the proof. $\qquad\square$

*Proof of Theorem C.4.* We first use induction to show that the following two conditions holds for all time steps $t < T$:

$$x_t \in \mathcal{B}\left(x_t^*, R\right), \tag{25a}$$

$$e_{t|mn} \leq \sum_{\tau=0}^{k} \left(\left(R_1 + D_{x^*}\right) \cdot q_1(\tau) + q_2(\tau)\right) \rho_{mn,\tau} + 2R_2 \left(\left(R_1 + D_{x^*}\right) \cdot q_1(k) + q_2(k)\right). \tag{25b}$$

At time step 0, equation 25a holds because $x_0 = x_0^*$, and equation 25b holds by lemma C.1 and the assumption on the terminal cost $F_k$ of $\mathsf{MPC}_k$.

Suppose equation 25a and equation 25b hold for all time steps $\tau < t$. For time step $t$, by the assumption on the prediction errors $\rho_{t,\tau}$ and prediction horizon $k$ in Theorem C.4, we know that $e_\tau \leq \frac{R}{C_3^2 L_g}$ holds for all $\tau < t$ because equation 25b holds for all $\tau < t$. Thus, we know that equation 25a holds for time step $t$ by Equation (21) since

$$\|x_t - x_t^*\| \leq L_* \sum_{i=0}^{n} q_3(t - (i+1)m) \sum_{\tau=mi}^{m(i+1)-1} e_{\tau|mi}$$

$$\leq mL_* \frac{R_1}{mC_3 L_*} \sum_{i=0}^{n} q_3(t - (i+1)m)$$

$$\leq \frac{R_1}{C_3} \cdot C_3 = R_1$$

Then, since equation 25a holds for time step $t$, and the terminal cost $F_{t+k}$ of $\mathsf{MPC}_k$ is set to be the indicator function of some state $\bar{y}(\xi_{t+k|t})$ that satisfies $\bar{y}(\xi_{t+k|t}) \in \mathcal{B}(x_{t+k}^*, R_1)$ if $t < T - k$, we know equation 25b also holds for time step $t$ by Lemma C.1. This finishes the induction proof of equation 25.

To simplify the notation, let $R_0 := R_1 + D_{x^*}$. Note that equation 25b implies that

$$e_{t|mn}^2 \leq \left(\sum_{\tau=0}^{k} \left(R_0 \cdot q_1(\tau) + q_2(\tau)\right) + 2R_2 \left(R_0 + 1\right)\right)$$

$$\cdot \left(\sum_{\tau=0}^{k} \left(R_0 \cdot q_1(\tau) + q_2(\tau)\right) \rho_{mn,\tau}^2 + 2R_2 \left(R_0 \cdot q_1(k)^2 + q_2(k)^2\right)\right) \tag{26a}$$

$$\leq \left(R_0 C_1 + C_2 + 2R_2(R_0 + 1)\right)$$

$$\cdot \left(\sum_{\tau=0}^{k-1} \left(R_0 \cdot q_1(\tau) + q_2(\tau)\right) \rho_{mn,\tau}^2 + \left(2R_2 + 1\right) \left(R_0 \cdot q_1(k)^2 + q_2(k)^2\right)\right), \tag{26b}$$

where we use the Cauchy-Schwarz inequality in equation 26a; we use the bounds $\sum_{\tau=0}^{k} q_1(\tau) \leq C_1$ and $\sum_{\tau=0}^{k} q_2(\tau) \leq C_2$ in equation 26b.

$$\sum_{n=0}^{N} \sum_{t=mn}^{m(n-1)-1} e_{t|mn}^2 \tag{27a}$$

$$\leq \left(R_0 C_1 + C_2 + 2R_2(R_0 + 1)\right)$$

$$\cdot \left(\sum_{\tau=0}^{k-1} \left(R_0 \cdot q_1(\tau) + q_2(\tau)\right) \sum_{n=0}^{N} \sum_{t=mn}^{m(n-1)-1} \rho_{mn,\tau}^2 + \left(2R_2 + 1\right) \left(R_0 \cdot q_1(k)^2 + q_2(k)^2\right) T\right)$$

$$\tag{27b}$$

$$\leq \left(R_0 C_1 + C_2 + 2R_2(R_0 + 1)\right)$$

$$\cdot \left(\sum_{\tau=0}^{k-1} \left(R_0 \cdot q_1(\tau) + q_2(\tau)\right) \sum_{n=0}^{N} m\rho_{mn,\tau}^2 + \left(2R_2 + 1\right) \left(R_0 \cdot q_1(k)^2 + q_2(k)^2\right) T\right) \tag{27c}$$

Since equation 25 and equation 29 holds for all time steps $t < T$, we can apply Lemma C.3 to obtain that

$$\text{cost}(\text{MPC}_k^m) - \text{cost}(\text{OPT}) = O\left(mL_*^2\sqrt{\text{cost}(\text{OPT}) \cdot E_1} + mL_*^2 E_1\right) ,$$

where

$$E_1 := (R_0 C_1 + C_2 + 2R_2(R_0 + 1))$$
$$\cdot \left(\sum_{\tau=0}^{k-1}(R_0 \cdot q_1(\tau) + q_2(\tau))\sum_{n=0}^{N}m\rho_{mn,\tau}^2 + (2R_2 + 1)\left(R_0 \cdot q_1(k)^2 + q_2(k)^2\right)T\right)$$

This finishes the proof of Theorem C.4. □

### D.2 PROOF FOR $\text{MPC}_{k,\epsilon}$

*Proof of Lemma C.5.* We bound the per-step error by decomposing it into two parts: the error due to state deviation from the planned trajectory, and the error due to parameter uncertainty.

Let $\hat{x}_{t|t'} := \psi_{t'}^{t'+k}(x_{t'}, \xi_{t':t'+k-1|t'}, \bar{y}(\xi_{t+k|t'})_{y_t}$ be the planned state at time $t$ from the plan computed at time $t'$. We have

$$e_t = \left\|\psi_{t'}^{t'+k}(x_{t'}, \xi_{t':t'+k-1|t}, \bar{y}(\xi_{t'+k|t'}); \mathbb{I})_{v_t} - \psi_t^T(x_t, \xi_{t:T}^*; F_T)_{v_t}\right\|$$

$$= \left\|\psi_t^{t'+k}(\hat{x}_{t|t'}, \xi_{t:t'+k-1|t'}, \bar{y}(\xi_{t'+k|t}); \mathbb{I})_{v_t} - \psi_t^{t'+k}(x_t, \xi_{t:t'+k-1}^*, x_{t'+k|t}^*; \mathbb{I})_{v_t}\right\| \quad (28a)$$

$$\leq \left\|\psi_t^{t'+k}(\hat{x}_{t|t'}, \xi_{t:t'+k-1|t'}, \bar{y}(\xi_{t'+k|t'}); \mathbb{I})_{v_t} - \psi_t^{t'+k}(x_t, \xi_{t:t'+k-1|t'}, \bar{y}(\xi_{t'+k|t'}); \mathbb{I})_{v_t}\right\|$$

$$+ \left\|\psi_t^{t'+k}(x_t, \xi_{t:t'+k-1|t'}, \bar{y}(\xi_{t'+k|t'}); \mathbb{I})_{v_t} - \psi_t^{t'+k}(x_t, \xi_{t:t'+k-1}^*, x_{t+k|t}^*; \mathbb{I})_{v_t}\right\|$$

$$\leq q_3(0)\left\|\hat{x}_{t|t'} - x_t\right\| \quad (28b)$$

$$+ \sum_{\tau=t-t'}^{t'+k-t-1}\left(\|x_t\| \cdot q_1(\tau) + q_2(\tau)\right)\rho_{t',\tau} + \left(\|x_t\| \cdot q_1(k) + q_2(k)\right)\left\|\bar{y}(\xi_{t+k|t}) - x_{t+k|t}^*\right\|$$
$$\quad (28c)$$

$$\leq q_3(0)\epsilon + \sum_{\tau=0}^{k-1}\left((D_{x^*} + R_1) \cdot q_1(\tau) + q_2(\tau)\right)\rho_{t',\tau} + 2R_2\left(\|x_t\| \cdot q_1(k) + q_2(k)\right). \quad (28d)$$

In equation 28a, we use the fact that the imagined optimal trajectory starting from imagined state $\hat{x}_{t|t'}$ (i.e. $\psi_t^{t'+k}(\hat{x}_{t|t'}, \xi_{t:t'+k-1|t}, \bar{y}(\xi_{t'+k|t}); \mathbb{I})_{y_{t:t'+k}, v_{t:t'+k}})$ is sub-trajectory of the imagined optimal trajectory starting from state $x_{t'}$(i.e. $\psi_{t'}^{t'+k}(x_{t'}, \xi_{t':t'+k-1|t}, \bar{y}(\xi_{t'+k|t'}); \mathbb{I})_{y_{t':t'+k}, v_{t':t'+k}})$. In equation 28b we apply perturbation bound equation 10 and in equation 28c we apply perturbation bound equation 11. Equation (28d) comes from the assumption that $x_t \in \mathcal{B}(x_t^*, R_1)$.

□

*Proof of Theorem C.6.* This proof is a simple extension of the proof of Theorem C.4 and Theorem 3.3 in (Lin et al., 2022).

By Lemma 3.2 in (Lin et al., 2022), we have

$$\|x_t - x_t^*\| \leq L\sum_{i=0}^{t-1}q_3(i)e_{t-i-1} \leq L \cdot \frac{R_1}{C_3 L}\sum_{i=0}^{t-1}q_3(i) \leq R_1 .$$

With the same induction as Theorem C.4 we can see $x_t \in \mathcal{B}(x_t^*, R_1)$ so the perturbation bounds hold.

To simplify the notation, let $R_0 := R_1 + D_{x^*}$. Note that equation 17 implies that

$$
\begin{aligned}
e_t^2 \leq{} & (q_3(0)\epsilon)^2 \\
& + \left( \sum_{\tau=0}^{k} (R_0 \cdot q_1(\tau) + q_2(\tau)) \, \rho_{p(t),\tau} + 2R_2 \left( R_0 \cdot q_1(k) + q_2(k) \right) \right)^2 \\
& + 2q_3(0)\epsilon \left( \sum_{\tau=0}^{k} (R_0 \cdot q_1(\tau) + q_2(\tau)) \, \rho_{p(t),\tau} + 2R_2 \left( R_0 \cdot q_1(k) + q_2(k) \right) \right) \quad \text{(29a)}
\end{aligned}
$$

$$
\begin{aligned}
\leq{} & q_3(0)^2 \epsilon^2 \\
& + \left( \sum_{\tau=0}^{k} (R_0 \cdot q_1(\tau) + q_2(\tau)) + 2R_2 \left( R_0 + 1 \right) \right) \\
& \quad \cdot \left( \sum_{\tau=0}^{k} (R_0 \cdot q_1(\tau) + q_2(\tau)) \, \rho_{p(t),\tau}^2 + 2R_2 \left( R_0 \cdot q_1(k)^2 + q_2(k)^2 \right) \right) \\
& + 2q_3(0)\epsilon \left( \sum_{\tau=0}^{k} (R_0 \cdot q_1(\tau) + q_2(\tau)) \, \rho_{p(t),\tau} + 2R_2 \left( R_0 \cdot q_1(k) + q_2(k) \right) \right) \quad \text{(29b)}
\end{aligned}
$$

$$
\begin{aligned}
\leq{} & O \left( \epsilon^2 + \epsilon \left( \sum_{\tau=0}^{k} (R_0 \cdot q_1(\tau) + q_2(\tau)) \, \rho_{p(t),\tau} + 2R_2 \left( R_0 \cdot q_1(k) + q_2(k) \right) \right) \right) \\
& + O \left( R_0 C_1 + C_2 + 2R_2(R_0 + 1) \right) \\
& \quad \cdot \left( \sum_{\tau=0}^{k-1} (R_0 \cdot q_1(\tau) + q_2(\tau)) \, \rho_{p(t),\tau}^2 + (2R_2 + 1) \left( R_0 \cdot q_1(k)^2 + q_2(k)^2 \right) \right), \quad \text{(29c)}
\end{aligned}
$$

where $p(t)$ denotes the last time step that plans before $t$. We use the Cauchy-Schwarz inequality in equation 29b; we use the bounds $\sum_{\tau=0}^{k} q_1(\tau) \leq C_1$, $\sum_{\tau=0}^{k} q_2(\tau) \leq C_2$ in equation 29c.

Finally, we apply Lemma 3.2 in (Lin et al., 2022) and Lemma F.2 in Lin et al. (2021) to connect the per-step errors to overall performance:

$$
\begin{aligned}
& \text{cost}(\mathsf{MPC}_{k,\epsilon}) - \text{cost}(\mathsf{OPT}) \\
& \leq \sqrt{ \left( \frac{\ell}{2} \cdot (1 + 2C_3 L^2) \cdot (1 + C_3) \right) \cdot \text{cost}(\mathsf{OPT}) \cdot \sum_{t=0}^{T-1} e_t^2 } \\
& \quad + \frac{\ell}{2} \cdot \left( 1 + 2C_3 L^2 \right) \cdot (1 + C_3) \cdot \sum_{t=0}^{T-1} e_t^2. \quad \text{(30a)} \\
& = O \left( \sqrt{L^2 \text{cost}(\mathsf{OPT}) \cdot (E_2 + {+}\epsilon E + \epsilon^2 T)} + L^2 (E_2 + \epsilon E + \epsilon^2 T) \right), \quad \text{(30b)}
\end{aligned}
$$

where

$$
\begin{aligned}
E_2 := {} & (R_0 C_1 + C_2 + 2R_2(R_0 + 1)) \\
& \cdot \left( \sum_{\tau=0}^{k-1} (R_0 \cdot q_1(\tau) + q_2(\tau)) \sum_{t=0}^{T} \rho_{p(t),\tau}^2 + (2R_2 + 1) \left( R_0 \cdot q_1(k)^2 + q_2(k)^2 \right) T \right).
\end{aligned}
$$

$\square$

### D.3 PROOF FOR $\mathsf{MPC}_{AR}$

*Proof.* We extend the analysis of $\mathsf{MPC}_{k,\epsilon}$ to handle time-varying thresholds. The key insight is that our adaptive threshold choice allows us to bound the additional regret terms more tightly.

First we introduce an adaptation of Equation (29) for time-varying thresholds.

$$e_t^2 \leq \left( q_3(0)\epsilon_t + \sum_{\tau=0}^{k} \left( R_0 \cdot q_1(\tau) + q_2(\tau) \right) \rho_{p(t),\tau} + 2R_2 \left( R_0 \cdot q_1(k) + q_2(k) \right) \right)^2 \tag{31a}$$

$$\leq \left( q_3(0)^2 \epsilon_t + \sum_{\tau=0}^{k} \left( R_0 \cdot q_1(\tau) + q_2(\tau) \right) + 2R_2 \left( R_0 + 1 \right) \right)$$

$$\cdot \left( \epsilon_t + \sum_{\tau=0}^{k} \left( R_0 \cdot q_1(\tau) + q_2(\tau) \right) \rho_{p(t),\tau}^2 + 2R_2 \left( R_0 \cdot q_1(k)^2 + q_2(k)^2 \right) \right) \tag{31b}$$

$$\leq \left( R_0 C_1 + C_2 + 2R_2(R_0 + 1) \right)$$

$$\cdot \left( \sum_{\tau=0}^{k-1} \left( R_0 \cdot q_1(\tau) + q_2(\tau) \right) \rho_{p(t),\tau}^2 + (2R_2 + 1) \left( R_0 \cdot q_1(k)^2 + q_2(k)^2 \right) \right)$$

$$+ O \left( \epsilon_t^2 + \epsilon_t \left( \sum_{\tau=0}^{k-1} \left( R_0 \cdot q_1(\tau) + q_2(\tau) \right) \rho_{p(t),\tau}^2 + (2R_2 + 1) \left( R_0 \cdot q_1(k)^2 + q_2(k)^2 \right) \right) \right),$$

$$\tag{31c}$$

where we use Cauchy-Schwarz inequality in Equation (31b).

Then we bound $\|x_t - x_t^*\|$ in a more fine-grained way to account for the adaptive nature of our algorithm:

$$\|x_t - x_t^*\| = \left\| x_t - \psi_0^T (x_0)_{y_t} \right\|$$

$$\leq \left\| x_t - \psi_{t-1}^T (x_{t-1})_{y_t} \right\| + \sum_{i=1}^{t-1} \left\| \psi_{t-i}^T (x_{t-i})_{y_t} - \psi_{t-i-1}^T (x_{t-i-1})_{y_t} \right\|$$

$$\leq \left\| x_t - \psi_{t-1}^T (x_{t-1})_{y_t} \right\| + \sum_{i=1}^{t-1} q_3(i) \left\| x_{t-i} - \psi_{t-i-1}^T (x_{t-i-1})_{y_{t-i}} \right\| \tag{32}$$

$$\leq \sum_{i=0}^{t-1} q_3(i) \left\| x_{t-i} - \psi_{t-i-1}^T (x_{t-i-1})_{y_{t-i}} \right\| \tag{33}$$

$$\leq \sum_{i=0}^{t-1} q_3(i) L_{t-i-1} e_{t-i-1}, \tag{34}$$

Taking squares and applying Cauchy-Schwarz:

$$\|x_t - x_t^*\|^2 \leq \left( \sum_{i=0}^{t-1} q_3(i) L_{t-i-1} e_{t-i-1} \right)^2$$

$$\leq \left( \sum_{i=0}^{t-1} q_3(i) \right) \cdot \left( \sum_{i=0}^{t-1} q_3(i) L_{t-i-1}^2 e_{t-i-1}^2 \right) \tag{35a}$$

$$\leq C_3 \left( \sum_{i=0}^{t-1} q_3(i) L_{t-i-1}^2 e_{t-i-1}^2 \right). \tag{35b}$$

Similarly, we can bound the control deviations:

$$\|u_t - u_t^*\|^2 \le \left(e_t + \sum_{i=0}^{t-1} q_3(i) L_{t-i-1} e_{t-i-1}\right)^2$$

$$\le \left(1 + \sum_{i=0}^{t-1} q_3(i)\right) \cdot \left(e_t^2 + \sum_{i=0}^{t-1} q_3(i) L_{t-i-1}^2 e_{t-i-1}^2\right) \tag{36a}$$

$$\le (1 + C_3) \cdot \left(\sum_{i=0}^{t} q_3(i) L_{t-i-1}^2 e_{t-i-1}^2\right), \tag{36b}$$

Summing equation 35 and equation 36 over time steps $t$ gives that

$$\sum_{t=1}^{T} \|x_t - x_t^*\|^2 + \sum_{t=0}^{T-1} \|u_t - u_t^*\|^2$$

$$\le C_3 \sum_{t=1}^{T} \left(\sum_{i=0}^{t-1} q_3(i) L_{t-i-1}^2 e_{t-i-1}^2\right) + (1 + C_3) \cdot \sum_{t=0}^{T-1} \left(\sum_{i=0}^{t-1} q_3(i) L_{t-i-1}^2 e_{t-i-1}^2\right)$$

$$\le (1 + 2C_3) \sum_{t=0}^{T-1} L_t^2 e_t^2, \tag{37}$$

where we rearrange the terms and use $\sum_{j=0}^{\infty} q_3(j) \le C_3$ in the last inequality.

By Lemma F.2 in Lin et al. (2021) with similar analysis in previous theorem, we arrive at

$$\mathrm{cost}(\mathsf{MPC}_{k,\epsilon}) - \mathrm{cost}(\mathsf{OPT})$$

$$\le \sqrt{\left(\frac{\ell}{2} \cdot (1 + 2C_3)\right) \cdot \mathrm{cost}(\mathsf{OPT}) \cdot \sum_{t=0}^{T-1} L_t^2 e_t^2} + \frac{\ell}{2} \cdot (1 + 2C_3) \cdot \sum_{t=0}^{T-1} L_t^2 e_t^2 \,.$$

Now we focus on $L_t^2 e_t^2$ and the three terms in Equation (31c). The first term corresponds to the regret of standard $\mathsf{MPC}_k^1$. The second term captures the effect of our adaptive threshold, which we show is well-controlled. To see this, we have

$$L_t \epsilon_t \le \epsilon_0 L_t \exp(-\alpha_L \widehat{L}_t) \le \epsilon_0 \frac{L_t}{e \cdot \alpha_L \cdot \widehat{L}_t} \le \epsilon_0 \frac{L_t}{e \cdot \alpha_L \cdot \lambda L_t} = \frac{\epsilon_0}{e \lambda \alpha_L} \,, \tag{38a}$$

where we use the inequality $\exp(-x) \le \frac{1}{ex}$ for $x > 0$ in the second step. This gives us

$$L_t^2 \epsilon_t^2 \le \frac{\epsilon_0^2}{e^2 \lambda^2 \alpha_L^2} \,. \tag{38b}$$

For the third term, we denote $s_t := \sum_{\tau=0}^{k-1} (R_0 \cdot q_1(\tau) + q_2(\tau)) \rho_{p(t),\tau}^2 + (2R_2 + 1) \left(R_0 \cdot q_1(k)^2 + q_2(k)^2\right)$ for the simplicity of notation, thus

$$L_t^2 \epsilon_t s_t \le \epsilon_0 \cdot \left(L_t^2 \exp(-\alpha_L L_t)\right) \cdot \left(\exp(-\alpha_\delta \delta_t^o) s_t\right) \tag{38c}$$

$$\le \epsilon_0 \cdot \left(\frac{4L_t^2}{e^2 \alpha_L^2 \widehat{L}_t^2}\right) \cdot \left(\frac{s_t}{e \alpha_\delta \delta_t^o}\right) \tag{38d}$$

$$\le \epsilon_0 \cdot \left(\frac{4}{e^2 \alpha_L^2 \lambda^2}\right) \cdot \left(\frac{1}{e \alpha_\delta \mu}\right) = \frac{4\epsilon_0}{e^3 \lambda^2 \mu \alpha_L^2 \alpha_\delta} \,, \tag{38e}$$

where we use the fact that $\exp(-x) \le \frac{1}{ex}$ and $\exp(-x) \le \frac{4}{e^2 x^2}$ in the second inequality. Combining everything together, we have our final regret bound

$$\text{cost}(\mathsf{MPC}_{k,\epsilon}) - \text{cost}(\mathsf{OPT})$$

$$\leq O\left(\sqrt{\text{cost}(\mathsf{OPT})(L^2 E + \frac{\epsilon_0}{\alpha_L^2}(\epsilon_0 + \frac{1}{\alpha_\delta})T)} + L^2 E + \frac{\epsilon_0}{\alpha_L^2}(\epsilon_0 + \frac{1}{\alpha_\delta})T\right).$$

where

$$E_2 \coloneqq (R_0 C_1 + C_2 + 2R_2(R_0 + 1))$$

$$\cdot \left(\sum_{\tau=0}^{k-1} (R_0 \cdot q_1(\tau) + q_2(\tau)) \sum_{t=0}^{T} \rho_{p(t),\tau}^2 + (2R_2 + 1)\left(R_0 \cdot q_1(k)^2 + q_2(k)^2\right)T\right).$$

The above result may not recover to previous results when $\alpha_L = \alpha_\delta = 0$. To mitigate this, we not need to apply the inequality $\exp(-x) \leq \frac{1}{1+x^2}$ and $\exp(-x) \leq \frac{1}{1+x^2}$ in Equation (38a) and Equation (38d) when $x$ are extremely small. So it can still be proved that the regret of $\mathsf{MPC}_{AR}$ is no more than regret of $\mathsf{MPC}_{k,\epsilon}$

$\square$

# E  EXPERIMENT DETAILS

This appendix provides further details on the experimental setup described in Section 4.

## E.1  SIMULATED EXPERIMENTS SETUP

Simulated experiments in this work are conducted on the VP2 Tian et al. (2023b) benchmark. All experiments can be conducted on a single NVIDIA GeForce RTX 4090 GPU. For each run of the experiments, it takes approximately 2500-3000 MiB of GPU memory for video prediction models like SVG (Villegas et al., 2019) and Struct-VRNN (Minderer et al., 2019). The simulator models and associated processes typically consume 20-25 GB of CPU RAM. Evaluating one learned model on a single task for one seed takes approximately 0.5-1 hour, while evaluations involving direct interaction with the simulator models (including noisy simulators in Appendix E.6 and visual vorrupted simulators in Appendix E.7 ) can take 2-3 hours per task per seed due to potentially different computational characteristics.

## E.2  REAL-WORLD EXPERIMENTS SETUP

As shown in Figure 9, our real-world experiments are conducted using a Franka Emika Panda robotic arm. The arm is equipped with a simple cubic pusher as its end-effector to interact with the environment and objects. To obtain precise and high-frequency measurements of the keypoints defining the state , we utilize a Vicon motion capture system. This system provides accurate 3D coordinate data, which forms the basis for the state representations $x_t$ used by our learned world models and the MPC controller in the real-world tasks.

## E.3  STATE-BASED WORLD MODELING FOR REAL-WORLD TASKS

For the real-world experiments utilizing state-based representations (as opposed to vision-based models used in simulation), we learn task-specific, single-step dynamics models $f$. These models predict the next state $x_{t+1}$ given the current state $x_t$ and the action $u_t$ applied by the robot's end-effector (pusher):

$$x_{t+1} = f(x_t, u_t). \tag{39}$$

Here, $u_t \in \mathbb{R}^2$ typically represents the 2D movement command sent to the end-effector. The specific state representation $x \in \mathbb{R}^d$ varies depending on the task:

- **Open Door Task:** The state $x \in \mathbb{R}^6$ comprises the 2D coordinates of the door hinge (axis) $(x_0, y_0)$, the door handle/endpoint $(x_1, y_1)$, and the robot end-effector (pusher) $(x_e, y_e)$, as

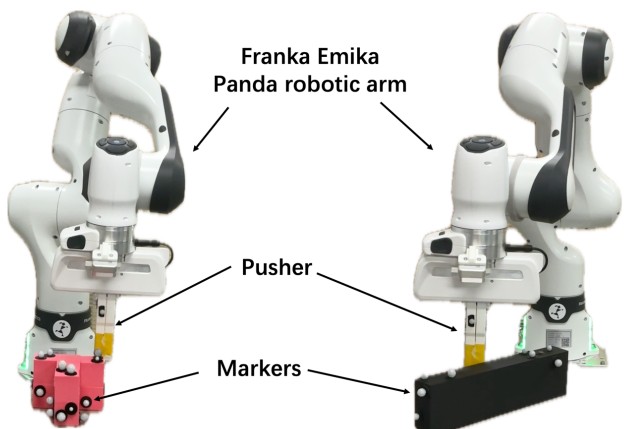

Figure 9: Real-World Experiments Setup

illustrated in Figure 10. Note that while the hinge position $(x_0, y_0)$ is fixed during any single task execution trial, its location may vary across different trials in the data collection phase. This variation encourages the learned model $f$ to generalize to different initial door configurations. We utilize classical 3 layer MLP model structure.

- **Push T-Block Task:** The state $x \in \mathbb{R}^8$ includes the 2D coordinates of three key points defining the T-block's pose (e.g., top-left $(x_1, y_1)$, top-right $(x_2, y_2)$, and bottom-middle $(x_3, y_3)$), along with the end-effector coordinates $(x_e, y_e)$, illustrated in Figure 11. These points allow tracking the object's position and orientation. We utilize classical 6 layer MLP model structure.

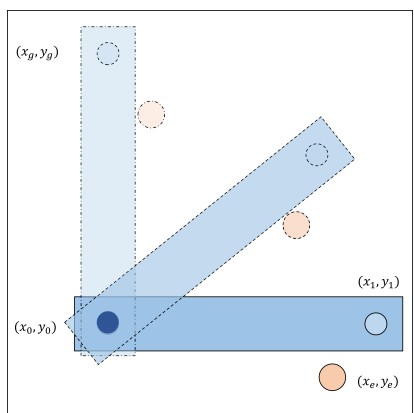

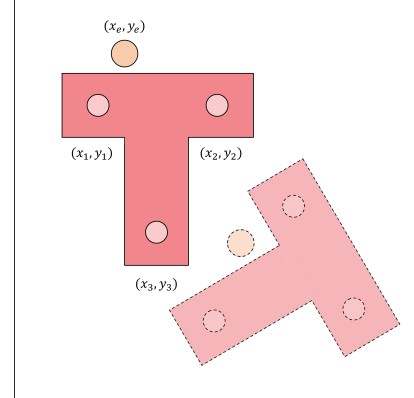

Figure 10: State representation for the Open Door task, showing hinge $(x_0, y_0)$, endpoint $(x_1, y_1)$, and pusher $(x_e, y_e)$.

Figure 11: State representation for the Push T-Block task, showing key points $(x_1, y_1)$, $(x_2, y_2)$, $(x_3, y_3)$ and pusher $(x_e, y_e)$.

### E.4 DATA COLLECTION FOR STATE-BASED MODELS

For each task setting requiring a state-based model, training data for the dynamics model $f$ (Eq. equation **??**) was collected from two primary sources:

1. **Expert Demonstrations:** A collection of successful task executions providing examples of effective interactions. Demonstrations were gathered via human teleoperation and potentially supplemented by trajectories generated using path planning algorithms (like RRT) to reach specific target configurations.

2. **Random Exploration:** Data gathered from trajectories generated by applying random actions within the operational workspace. This source contributes a significant portion of the dataset to ensure broad coverage.

Combining targeted expert data with broad exploration data aims to prevent the model $f$ from overfitting to specific demonstration trajectories. This strategy helps ensure the model captures dynamics across a wider range of the state-action space, potentially improving prediction accuracy along near-optimal paths discovered during online planning.

### E.5   WORLD MODEL TRAINING

The learned world models are trained as one-step predictors. For models trained from scratch (e.g., state-based models), we used the Adam optimizer with a learning rate of $5 \times 10^{-6}$ and a batch size of 16. These models were trained for a total of 300 epochs. During the inference phase for planning, multi-step future predictions are generated by recursively applying the learned one-step predictor. Specifically, to predict $k$ future steps given a sequence of actions from time $t$ to $t + k - 1$, the model autoregressively predicts $x_{t+1}, x_{t+2}, \ldots, x_{t+k}$.

### E.6   DETAILS FOR DISTURBED SIMULATORS AS WORLD MODEL

We add gaussian noise with mean $0$ and different std for different components with various disturbance level to construct noisy simulator, with details summarized in Table 3.

Table 3: Standard deviations (std) of Gaussian noise applied to state components for different disturbance levels in the noisy simulator experiments.

|  | Level 1 | Level 2 | Level 3 |
|---|---|---|---|
| robot position | 0.001 | 0.005 | 0.010 |
| robot velocity | 0.001 | 0.005 | 0.010 |
| object position | 0.000 | 0.001 | 0.005 |
| object velocity | 0.000 | 0.001 | 0.005 |
| end effector position | 0.001 | 0.005 | 0.010 |

### E.7   DETAILS FOR VISUAL CORRUPTED SIMULATORS AS WORLD MODEL

To assess the impact of imperfect visual predictions from a world model (distinct from errors in an underlying state representation), we apply two types of visual corruption to the images generated by an otherwise accurate simulator before they are processed (e.g., by a feature extractor like DINO):

1. **Gaussian Noise:** Additive Gaussian noise is applied directly to the pixel values of the predicted images to make them appear more noisy.

2. **Gaussian Blur:** A Gaussian filter is applied to the predicted images to make them appear more blurry.

The parameters for these visual corruptions are detailed in Tables 4 and 5 and visualizations are in Figures 12 and 13.

Table 4: Parameters for adding Gaussian noise to predicted images. 'std' refers to the standard deviation of the noise.

| Noise Level | std |
|---|---|
| Level 1 | 0.1 |
| Level 2 | 0.2 |
| Level 3 | 0.5 |

Table 5: Parameters for applying Gaussian blur to predicted images. 'sigma' refers to the standard deviation (kernel size) of the Gaussian filter.

| Blurriness Level | sigma |
|---|---|
| Level 1 | 0.1 |
| Level 2 | 0.5 |
| Level 3 | 1.0 |

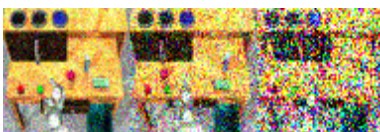

Figure 12: Visualization of different Noise Levels.

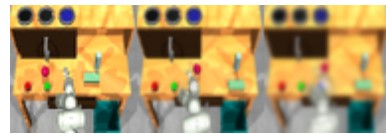

Figure 13: Visualization of different Blurriness Levels.

# F MORE EXPERIMENTAL RESULTS

## F.1 EXPERIMENTAL RESULTS FOR $\text{MPC}_k^m$ AND $\text{MPC}_{k,\epsilon}$

We first explore how fixed $m$ (number of steps to execute per plan) and $\epsilon$ (deviation threshold for replanning) influence the performance (success rate) and computational cost (NFE) of world model planning in simulation. This provides baseline intuition for understanding the trade-offs involved and motivates the need for adaptive approaches. The results are presented for the SVG and Struct-VRNN models on the VP2 benchmark tasks.

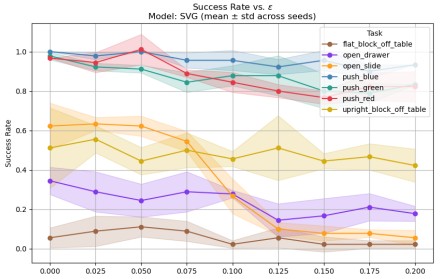
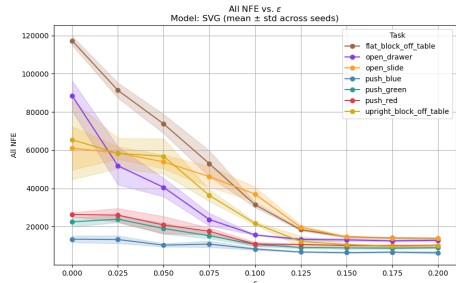

Figure 14: Performance of SVG model with $\text{MPC}_{k,\epsilon}$ for different deviation thresholds $\epsilon$.

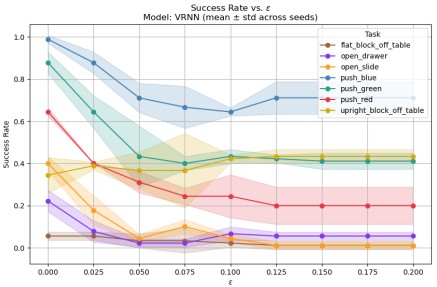
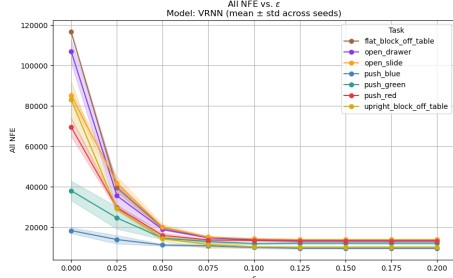

Figure 15: Performance of Struct-VRNN model with $\text{MPC}_{k,\epsilon}$ for different deviation thresholds $\epsilon$.

As our theoretical analysis suggests (Section 3.1), both task performance (success rate) and computational cost (NFE) tend to decrease as $m$ (number of steps per plan) and $\epsilon$ (replanning threshold) increase. However, the rate at which performance degrades varies significantly. This variation is influenced by factors such as the prediction quality of the specific world model and the sensitivity of the local dynamics encountered along the trajectories across different tasks. These results highlight the challenge of selecting a single fixed $m$ or $\epsilon$ that performs optimally across all scenarios, motivating our adaptive approach.

## F.2 TRAJECTORY VISUALIZATION OF REAL-WORLD EXPERIMENTS

The trajectory visualizations of our real-world experiments, presented in Figures 18 to 22, demonstrate the significant advantages of our adaptive replanning algorithm (ADAREP). These figures qualitatively

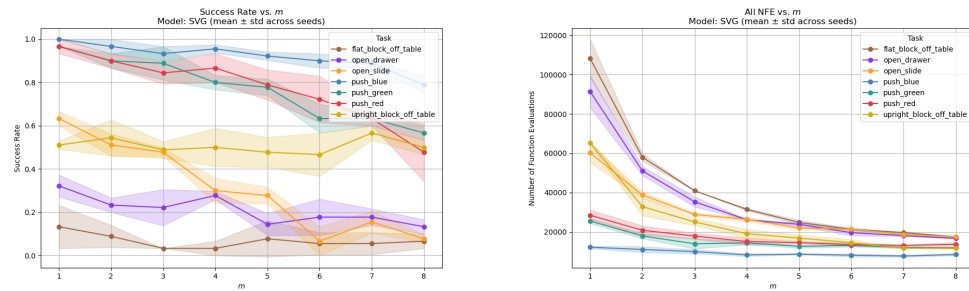

Figure 16: Performance of SVG model with $\mathrm{MPC}_k^m$ for different numbers of executed steps $m$.

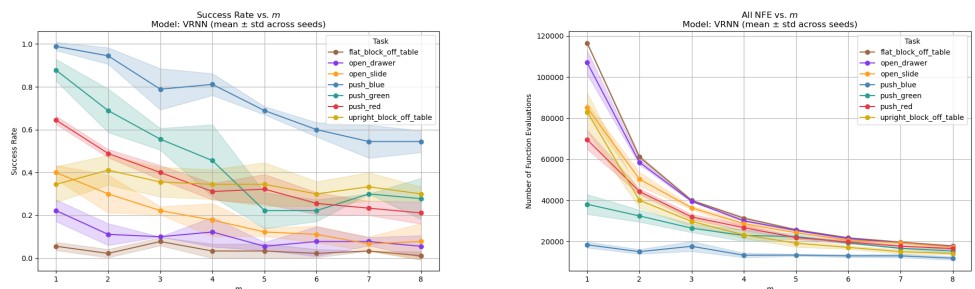

Figure 17: Performance of Struct-VRNN model with $\mathrm{MPC}_k^m$ for different numbers of executed steps $m$.

compare the behavior of ADAREP against the baseline $\mathrm{MPC}_k^1$. Different colors along an agent's path indicate segments executed from distinct plans; consequently, fewer color changes in the trajectories generated by ADAREP visually highlight its reduced replanning frequency and the associated computational savings.

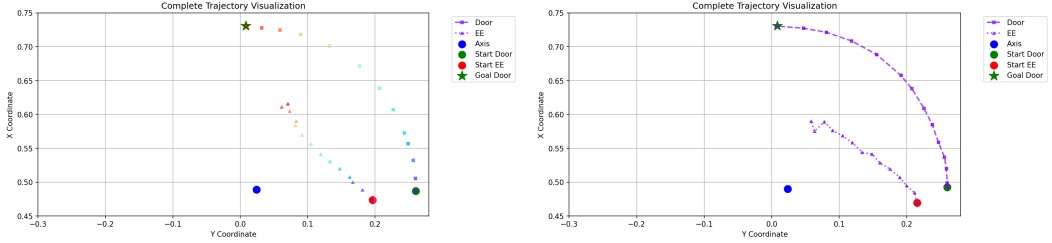

Figure 18: Trajectory of opening door to $90°$. Left: $\mathrm{MPC}_k^1$. Right: $\mathrm{MPC}_{AR}$.

# G  MORE ANALYSIS AND DISCUSSIONS

ADAREP does require additional hyperparameter tuning, how ever we clarify that the tuning for ADAREP is a modest, one-time, upfront cost that yields substantial, continuous benefits.

Our tuning process is efficient as it builds upon the baseline. A practitioner can start with a reasonable fixed threshold $\epsilon$ and then simply tune ADAREP's parameters , which intuitively control the adaptation's sensitivity. This is significantly faster than exhaustively searching for the "perfect".

We want to further demonstrate that both our adaptive method (ADAREP) and other non-adaptive method ($\mathrm{MPC}_k^m$, $\mathrm{MPC}_{k,\epsilon}$) rely on hyperparameter tuning, adaptive method is indeed much better.

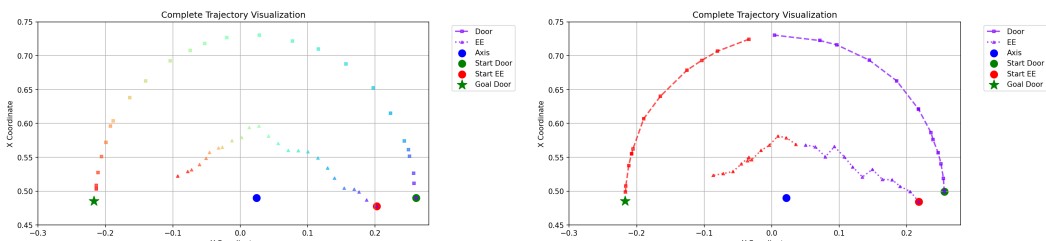

Figure 19: Trajectory of opening door to $180°$. Left: $\text{MPC}_k^1$. Right: $\text{MPC}_{AR}$. Different colors represent controls from different plans.

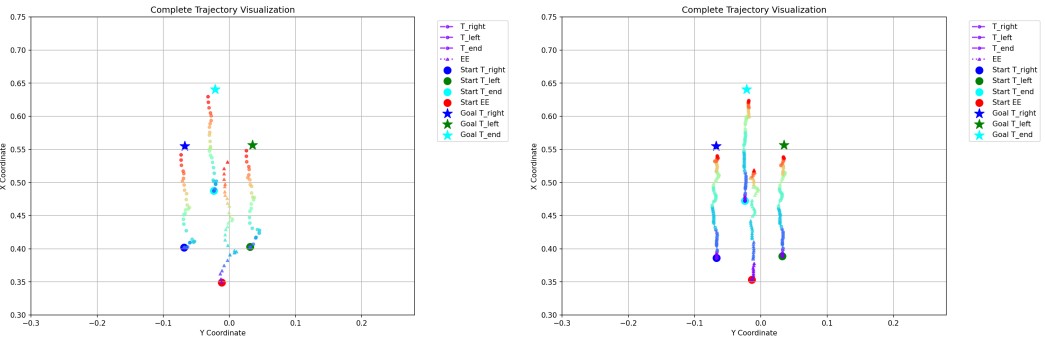

Figure 20: Trajectory of translating T-block. Left: $\text{MPC}_k^1$. Right: $\text{MPC}_{AR}$. Different colors represent controls from different plans.

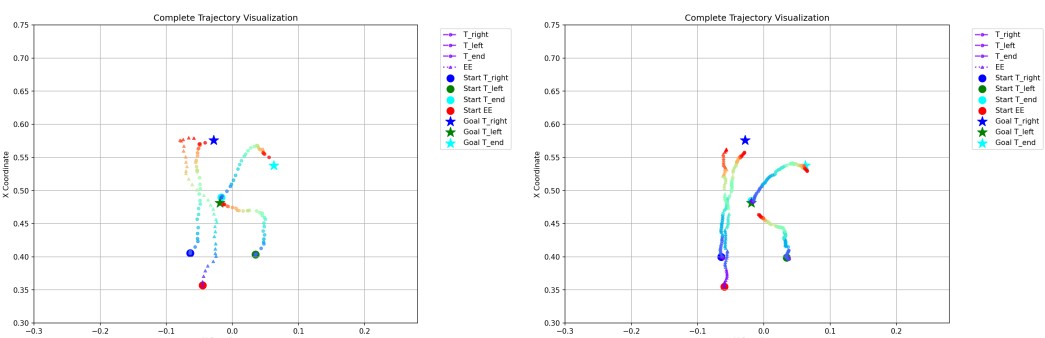

Figure 21: Trajectory of rotating T-block. Left: $\text{MPC}_k^1$. Right: $\text{MPC}_{AR}$. Different colors represent controls from different plans.

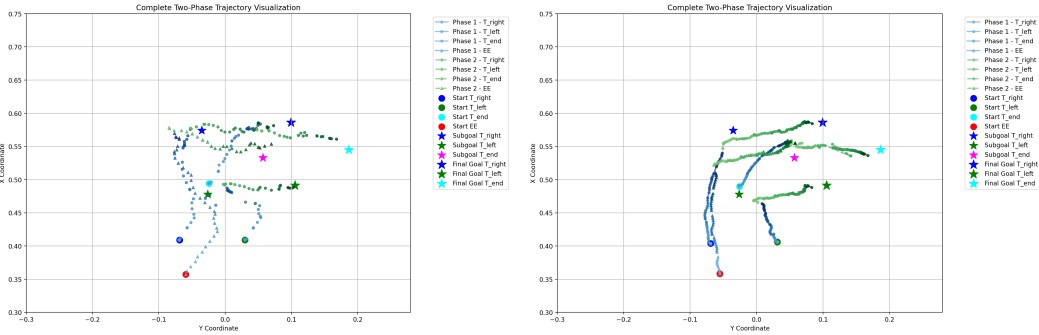

Figure 22: Trajectory of translating and rotating T-block. Left: $\text{MPC}_k^1$. Right: $\text{MPC}_{AR}$. Different colors represent controls from different plans.

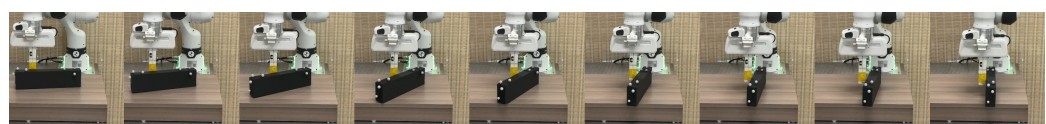

Figure 23: Visual demonstration for opening door to $90°$.

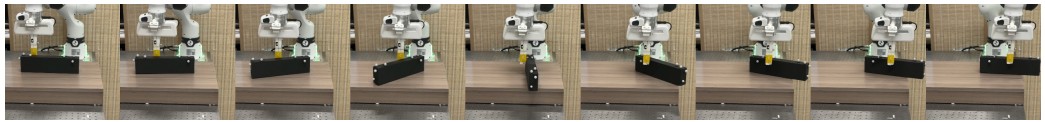

Figure 24: Visual demonstration for opening door to $180°$.

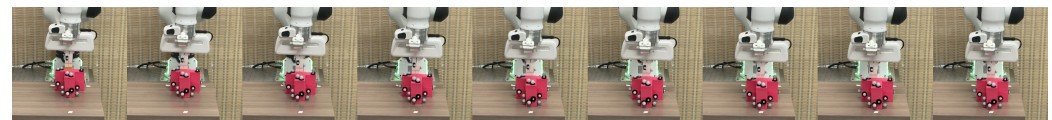

Figure 25: Visual demonstration for translating T-block.

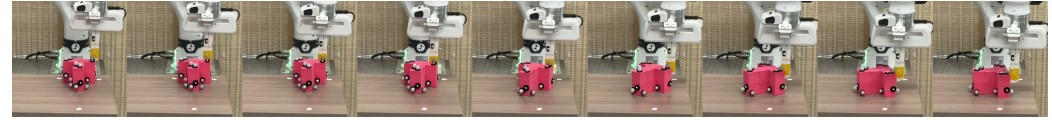

Figure 26: Visual demonstration for rotating T-block.

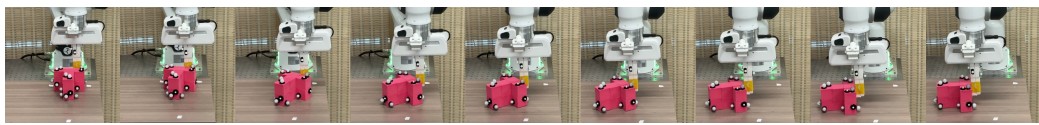

Figure 27: Visual demonstration for translating and rotating T-block.

This small upfront investment unlocks the key benefits of our method: significant NFE reductions at runtime, robust performance across diverse tasks without re-tuning, and the ability to adapt to changing dynamics within a single episode. We believe this favorable trade-off is a core strength of our approach.

## H    LIMITATIONS

Despite these promising results, this work has several limitations. A key challenge is the performance dependency on the quality of sensory input, particularly for vision-based models. As noted, future work should focus on developing more robust visual features, as the current approach's effectiveness can be hampered by visual corruptions or less discriminative features, leading to a performance gap compared to state-based world models where ADAREP excels. The current adaptive mechanism, while training-free and broadly applicable, relies on heuristic-driven parameter adjustments based on online estimates; the accuracy and reliability of these estimates are crucial, and noisy estimates could lead to suboptimal replanning decisions.

Future work should therefore focus on developing more robust visual features for vision-based models to broaden ADAREP's applicability. Additionally, exploring the integration of learning-based methods to further refine the adaptive parameters of ADAREP presents another promising research avenue, potentially leading to enhanced generalization and finer-grained adaptation across a wider array of tasks and conditions.

