# OpenReview forum: "AdaReP: Plug-and-Play Acceleration for World Model Predictive Control using Adaptive Re-Planning"
_ICLR.cc/2026/Conference — ICLR 2026 Conference Withdrawn Submission_

### Official Review · Reviewer_fE8h · 2025-10-30

**Soundness:** 3
**Presentation:** 4
**Contribution:** 2
**Rating:** 4
**Confidence:** 4

**Summary:**

The paper proposed an approach to adapt the replanning frequency during model predictive control with a world model. First the paper motivates the tradeoff between performance and computational cost and explain how existing methods strike different balances with these 2 variables and provide a theoretical analysis of the components contributing to the computational cost. With this knowledge the paper proposes a simple algorithm that can be plugged into any MPC framework which adapts the planning frequency based on 2 main criteria, 1) world model prediction error and 2) local dynamic sensitivity. The proposed approach is compared to multiple baselines in simulated and real-world environments. The results illustrate an improvement in the computational cost (based on the considered metric) while retaining performance on-par with frequently replanning MPC.

**Strengths:**

- The paper is very well written and enjoyable to read. I like the theoretical analysis as a motivation of the proposed approach
- The proposed approach makes sense, is elegantly simple, and can be plugged on top existing methods (at least for state-based world models)
- The results look quite promising, and are sufficient to support the central claim of the paper of reducing computational cost without strongly huring task performance
- Ablations show the role of different components and hyperparameters

**Weaknesses:**

- The choice of NFE as metric is a bit weak. Up to a certain limit, it's possible to parallelize world model rollouts, so only non-parallelizable aspects such as the number of outer loop CEM runs should count. It would also be very useful to provide computational time in seconds as a metric. I understand that the latter could vary with hardware, but it should still be interesting for readers and a good way to understand the impact of the method
- Latent state errors used in both the prediction error and the sensitivity are quite an uncalibrated value. I appreciate the authors honestly mentioning and illustrating this in their experiments section. However, this seems like a central limitation of the method. Given that the methodological contribution of the paper is quite simple, working out this detail would have substantially strenghened the contribution. As it stands, the method seems to be only reliably applicable to state-based systems.
- Some results are hard to interpret. For instance, in figure 2, in the top subfigures, less replanning with MPC outperforms regular MPC on the upright block task. This might be an outlier, which would suggest that the authors did not marginalize noise out of their evaluations (small amount of runs). I invite the authors to discuss such occurences in the main body of the paper and provide hypotheses or evidence supported explanation.



Minor issues:
- typo at Figure 2's caption: compuational --> computational
- typo at line 358: compuation --> computation

**Questions:**

- can the authors explain how less replanning can outperform more frequent planning?
- did the authors experiment with different estimators to be used to determine the replanning variable? if not, can you discuss your intuition about how to possibly solve this problem for latent world models in vision-based environments?

---

### Official Review · Reviewer_qyxd · 2025-10-31

**Soundness:** 3
**Presentation:** 2
**Contribution:** 2
**Rating:** 4
**Confidence:** 3

**Summary:**

MPC with world models requires frequent replanning that can be computationally expensive when using large world models. This paper proposes a method to replan less often, thereby alleviating these high computational needs. The replanning criterion is adaptive as a function of the model's prediction error and a measure of dynamics sensitivity. A theoretical analysis for the regret of the algorithm when replanning less frequently is provided. Results show that the proposed method reduces computations while maintaining performance.

**Strengths:**

The paper addresses an important problem, as planning with large world models can indeed be a computational burden. When analyzing the effect of replanning less often, the derived results correctly highlight dependencies on model accuracy and on the true system's dynamics. The experimental results include real-world experiments in robotic manipulation. Implementation details are clear. The limitations of the method for vision-based problems are clearly provided and hint at interesting future work.

**Weaknesses:**

1) The motivation and problem framing around replanning less frequently should be clarified. I see the benefit from an energy savings standpoint, but not for task performance, where one should replan as frequently as possible. The title (and other sentences) mentions *acceleration*, but it's unclear how replanning less often accelerates the planning process. Multiple sentences are similarly unclear:
- Lines 45-47: The statement "This frequent replanning (...) leading to (...) reduced control frequency due to delays." is unclear to me. Frequent replanning should instead *increase* control frequency.
- Line 184: The sentence "Natural approaches to accelerate the planning process involve reducing the replanning frequency." is unclear. Replanning less frequently does not mean solving the planning problem faster.

2) Weak discussion of related work: The paper correctly cites relevant works on world models, MPC, the NFE metric, and regret analysis. However, there is no related work section and no discussion of works studying the effect of replanning less often or with a shorter horizon to reduce model calls for computational savings, which is the crux of the contribution. The paper's analysis on MPC applies beyond world models, so one can expect relevant work in the literature (e.g., on event-triggered MPC and MPC with adaptive horizons).

3) Mathematical results can be improved:
- The assumptions, results statements, and proofs are in the appendix. The body of the paper could better discuss assumptions and results: a) How much do the proofs depart from those in previous work (are they trivial to derive?) and what are important and interesting steps (a proof sketch?), b) how tight are the bounds and could they be improved, and c) what are and how strong are the assumptions (e.g., convexity)?
- The proofs in the appendix are not polished and have some flaws. For example: 1) Lines 870, 874: $0<\lambda<0$ and $0<\mu<0$, 2) Theorem C.6 reports an inequality for $MPC_{k,\epsilon}$ with $m$ although $m$ is not a parameter of $MPC_{k,\epsilon}$, 3) Theorem C.7 reports bounds for MPCAR but the statement, inequality, and proof are for $MPC_{k,\epsilon}$, 4) the terminal cost is assumed smooth and convex, but is later set to an indicator function in Theorems C.6 and C.7, 5) In Property C.1., the following statement is vague and uses an undefined quantity: "assuming the underlying parameter sets $\Xi_t$ contain the relevant parameters:".

4) The algorithmic choice for the local dynamics sensitivity estimator is not well motivated. The metric $\|x_{t+1}-x_t\| / \|u_t\|$ is ill-posed if $\|u_t\|=0$, which could correspond to sensible actions. The practical algorithm uses $\|x_{t+1}-x_t\| / (\|u_t\| + \epsilon)$ in (8), which hints at better metrics that could be used, like the norm of the Jacobian of the dynamics.

**Questions:**

- How does the proposed method "accelerate the planning process", versus only requiring fewer model evaluation for computational savings?

- Local dynamics sensitivity estimator: Have you considered other metrics than the one in (5), and if so, what are the tradeoffs?

- Small clarification in Section E.4: The content and amount of demonstration could be described more precisely. The sentence "Demonstrations were (...) potentially supplemented by trajectories generated using path planning algorithms (like RRT)" is vague.

- Potentially interesting future work: How difficult would it be to extend your results to the case where one has access to a slow but accurate, and a fast to evaluate but less accurate prediction model? Being able to adaptively switch between the two using an extension of your method could open the door to acceleration, better performance, and computational savings.

Typos: 1) Line 356: "such the" => "such that the", 2) Line 392: "compuational", 3) Line 1508: "equation ??""

---

### Official Review · Reviewer_AUqf · 2025-10-31

**Soundness:** 3
**Presentation:** 3
**Contribution:** 3
**Rating:** 6
**Confidence:** 2

**Summary:**

Model Predictive Control (MPC) is a method for using World Models (WM) to plan in simulation in order to achieve some goal, measured by minimizing the cost function.  MPC works by evaluating the next $k$ steps into the future and looking for the actions which are the lowest cost in that timeframe.

However, WMs are in practice are not fully accurate, and also over time a drift develops between the true state of the world and the rollout in the world model when given the same actions, leading to an accumulation of errors.  This leads to sub-optimal or incorrect actions being taken which appear to work well in the WM but not in the simulator.  This is quantified by the dynamic regret cost - the difference in cost due to using a simulator with the wrong parameters.

The most common strategy for dealing with this drift is $MPC_k^m$, which builds a plan over $k$ steps but then only takes the first $m$ steps of it, discarding the rest. Another alternative is $MPC_{k,\delta}$, where $\delta$ is the max allowed difference between simulator and real world observations.

$MPC_k^1$ is the upper bound of performance (e.g., lowest regret) as it fully replans every step, but correspondingly also has the highest computational cost.   $MPC_k^m$ with $m>1$ aand  $MPC_{k,\delta}$ can decrease the computational demands by not as  frequently replanning the same of actions as it is not always required and the earlier plan steps would have still worked in practice.

The authors introduce a new dynamic regret method $MPC_{AR}$ that aims to have a better trade-off between the amount of replanning required and the regret induced due to not replanning. In addition to the model error, and unlike the baselines $MPC_k^m$ and $MPC_{k,\delta}$, the new metric incorporates the local sensitivity of the model to the actions, as that can be a indication that the noise is likely to make actions later in the plan less accurate.

While $MPC_{AR}$  has its own hyperparameters to tune ( $\alpha_L$, $\alpha_\eta$, and window size), the authors claim that parameters are easier to tune than the corresponding parameters in  $MPC_k^1$ and  $MPC_{k,\delta}$, and exhibit better transfer across tasks.

They then go on to derive dynamic regret bounds for the different versions of the MPC algorithm, proving that their method results in tighter bounds on the dynamic regret than $MPC_k^m$ and $MPC_{k,\eta}$.

They go on to empirically demonstrate that with settings to match the performance of $MPC_1^k}, they can save significant amounts of computation (around 20-90% ish percent).  This is Figure 2 in simulation, where they use a video world model, Figure 3 in a real world test, where they focus on the state representation derived using 3d keypoints and learn a one step world model.  They also address practical considerations, showing the sliding window for the error estimator is important.

**Strengths:**

# Originality
This is to my knowledge a new and novel adaptive bound for MPC, with a corresponding theoretical justification.

# Quality
The research was of high quality, where the algorithm is justified theoretically as well as empirically, and the analysis could lead to further innovations on how the bound can be tightened.   They clearly demonstrate the expected replanning behavior where pushing a door closer to the hinge does make the model replan more often, which does make sense.

# Clarity
In general the paper was clear and easy to follow and clear enough that the results could be reproduce from the descriptions within.  One minor point is that it wasn't clear on the first reading how $\delta_t$ was actually used downstream, as both L and E were used to compute it.

# Significance
MPC is heavily used in practice, and world models are getting significantly more expensive, like diffusion based video models. This makes saving rollouts becomes significantly more important, potentially increasing the applicability.  The applicability to video models also broadens the scope of where it can be applied.

**Weaknesses:**

There is a lot of richness to the underlying problem that is abstracted away - many prediction errors might not be relevant for the cost or the trajectory planning, and some are more important than others.  However, it should be said that dealing with the problem at this level allows their method to be widely applicable, making it a good baseline and allowing them to prove their regret bounds for their new method.

**Questions:**

Did you consider treating different parts of the prediction error differently?   For example not all prediction errors are relevant for the cost of the trajectory.  This would in particular be a problem if there was an background changes or activity in the video stream, which would be irrelevant for the task at hand but would drive a very frequent retraining signal.  The tradeoff is of course that this makes the method more widely applicable.

---

### Official Review · Reviewer_C6Gf · 2025-11-01

**Soundness:** 2
**Presentation:** 2
**Contribution:** 1
**Rating:** 4
**Confidence:** 4

**Summary:**

ADAREP is an adaptive re-planning rule that modulates Model-Predictive-Control (MPC) query frequency on-the-fly by shrinking or growing a drift-threshold ε based on **recent prediction-error and local dynamics sensitivity**.


ADAREP is positioned as a plug-and-play wrapper that can sit atop any learned world-model + MPC solver; experiments show large reductions up to 80% in model queries compared to replan-every-step MPC, while preserving task success on a set of simulated and real-robot manipulation benchmarks. The paper also analyses dynamic-regret and provides finite-time performance bounds.

**Strengths:**

## Main Results
- Evaluation in VP2 benchmark, 7 short-horizon manipulation tasks with two legacy video models, SVG and Struct-VRNN, under CEM and MPPI planners.
- Trials on a Franka Panda arm: “Open-Door 90°/180°” and “Push T-Block” variants, using a state-based world model.

## Strengths
- Simple heuristic that needs only a few lines on top of any MPC loop.
- Planning speed ups (10 × fewer model queries) on RoboDesk simulation and a Franka arm, with no drop in success rate.
- Reducing expensive world-model calls is a real bottleneck for planning. This work targets an important problem,

**Weaknesses:**

Here are some of Weaknesses :

- Incremental novelty: Deviation-triggered re-planning and Lipschitz-style sensitivity are long-standing ideas in robust MPC and already appear in recent RL planners such as TD-MPC [1] and the exploration schedules of Dreamer [2], while formal sensitivity decay is well studied.

- Weak benchmarks: Evaluation is limited to short-horizon, low-complexity tasks. More challenging suites like DMControl [3], OGBench [4], and Robomimic [5] are not included, so generality remains unclear.

- Out-dated model stack: Experiments use legacy video models SVG [6] and Struct-VRNN [7] with basic CEM/MPPI planners, omitting modern latent world models and planners such as PlaNet [8], Dreamer V4 [9], DIAMOND [10], TD-MPC [1], and DINO-WM [11].

References

[1] N. Hansen, X. Wang, and H. Su, “Temporal Difference Learning for Model Predictive Control,” ICLR, 2022. https://arxiv.org/abs/2203.04955


[2] D. Hafner, T. Lillicrap, J. Ba, and M. Norouzi, “Dream to Control: Learning Behaviors by Latent Imagination,” ICLR, 2020. https://arxiv.org/abs/1912.01603


[3] Y. Tassa et al., “DeepMind Control Suite,” arXiv:1801.00690, 2018.


[4] S. H. Park et al., “OGBench: Benchmarking Offline Goal-Conditioned RL,” NeurIPS, 2024. https://openreview.net/forum?id=M992mjgKzI


[5] A. Mandlekar et al., “robomimic: A Framework and Large-Scale Dataset for Robot Learning from Demonstration,” CoRL, 2021. https://arxiv.org/abs/2108.03298


[6] E. L. Denton and R. Fergus, “Stochastic Video Generation with a Learned Prior,” ICML, 2018. https://arxiv.org/abs/1802.07687



[7] M. Minderer et al., “Unsupervised Learning of Object Structure and Dynamics from Videos,” NeurIPS, 2019. https://arxiv.org/abs/1906.07889


[8] D. Hafner, T. Lillicrap, J. Ba, and M. Norouzi, “Learning Latent Dynamics for Planning from Pixels,” ICML, 2019. https://proceedings.mlr.press/v97/hafner19a.html


[9] D. Hafner et al., “Mastering Diverse Control Tasks through World Models,” Nature, 2025. https://doi.org/10.1038/s41586-025-08744-2


[10] E. Alonso et al., “Diffusion for World Modeling: Visual Details Matter in Atari,” NeurIPS, 2024. https://arxiv.org/abs/2405.12399


[11] G. Zhou et al., “DINO-WM: World Models on Pre-trained Visual Features Enable Zero-shot Planning,” arXiv:2411.04983, 2024.

**Questions:**

Questions:

- Could you report results on at least one high-dimensional continuous control (e.g. DMControl) be provided to support generality?
- How would ADAREP interact with uncertainty-aware planners that already modulate rollout depth (e.g. TD-MPC)?
- How would ADAREP interact with bare-bone world models like PlaNet or JEPA?
- Most of methods used in paper are backed by theoretical MPC framework. Could you please elaborate on assumptions to make the material more beginner friendly.


Overall, the method is practical yet largely composed of techniques already common in control and RL. The empirical section showcases efficiency on easy tasks, but stronger benchmarks and comparison to state-of-the-art world models would enhance the contribution.

---

### Note · Authors · 2026-01-06

**Comment:**

We have decided to withdraw our submission from ICLR 2026. We would like to sincerely thank the reviewers and the Area Chair for their time and valuable advice. We plan to use this feedback to improve our work and seek other opportunities for publication.

**Withdrawal Confirmation:**

I have read and agree with the venue's withdrawal policy on behalf of myself and my co-authors.